# PARP14 inhibition restores PD-1 immune checkpoint inhibitor response following IFNγ-driven acquired resistance in preclinical cancer models

Chun Wai Wong [1,2], Christos Evangelou [1,2], Kieran N. Sefton[1,2,8], Rotem Leshem [1,2,8], Wei Zhang [1], Vishaka Gopalan[3], Sorayut Chattrakarn [1,2], Macarena Lucia Fernandez Carro [1,2], Erez Uzuner [1,2], Holly Mole[1], Daniel J. Wilcock [1], Michael P. Smith [1], Kleita Sergiou[1], Brian A. Telfer [1], Dervla T. Isaac [4], Chang Liu[4], Nicholas R. Perl[4], Kerrie Marie [1], Paul Lorigan [1,5], Kaye J. Williams[1], Patricia E. Rao[6], Raghavendar T. Nagaraju [1,7], Mario Niepel [4] & Adam F. L. Hurlstone [1,2] ✉

Resistance mechanisms to immune checkpoint blockade therapy (ICBT) limit its response duration and magnitude. Paradoxically, Interferon γ (IFNγ), a key cytokine for cellular immunity, can promote ICBT resistance. Using syngeneic mouse tumour models, we confirm that chronic IFNγ exposure confers resistance to immunotherapy targeting PD-1 (α-PD-1) in immunocompetent female mice. We observe upregulation of poly-ADP ribosyl polymerase 14 (PARP14) in chronic IFNγ-treated cancer cell models, in patient melanoma with elevated *IFNG* expression, and in melanoma cell cultures from ICBT-progressing lesions characterised by elevated IFNγ signalling. Effector T cell infiltration is enhanced in tumours derived from cells pre-treated with IFNγ in immunocompetent female mice when PARP14 is pharmacologically inhibited or knocked down, while the presence of regulatory T cells is decreased, leading to restoration of α-PD-1 sensitivity. Finally, we determine that tumours which spontaneously relapse in immunocompetent female mice following α-PD-1 therapy upregulate IFNγ signalling and can also be re-sensitised upon receiving PARP14 inhibitor treatment, establishing PARP14 as an actionable target to reverse IFNγ-driven ICBT resistance.

Programmed cell death protein 1 (PD-1) is an immune checkpoint protein highly expressed on activated tumour-infiltrating T lymphocytes. Interaction with its ligand, programmed death ligand-1 (PD-L1), expressed on transformed, stromal and myeloid-derived cells in the tumour microenvironment (TME) and tumour-draining lymph nodes, promotes tumour immune evasion—and thereby disease progression— by suppressing effector T cell proliferation, migration, and anti-tumour immune responses while enhancing immune regulatory cells[1]. Therapeutics impeding PD-1/PD-L1 interactions and thereby antagonising PD-1 functions, have revolutionised the treatment of melanoma and other solid cancers such as non-small cell lung cancer (NSCLC), bladder carcinoma, and microsatellite instability-high (MSI-H) cancers[2].

The effectiveness of immune checkpoint blockade therapy (ICBT) is limited by multiple resistance mechanisms. While primary resistance

is widespread, cases where tumours initially respond but subsequently relapse within months or years, so-called acquired resistance, are also common[3]. Mechanisms of ICBT resistance are multifaceted and incompletely understood. As a key component of inflammatory signalling that characterises the TME, the cytokine interferon γ (IFNγ) exerts divergent effects on tumour immune responses, including those elicited by ICBT. Its role in promoting tumour immuno-surveillance is well established[4], and targets of IFNγ signalling are robust biomarkers of clinical response to ICBT[5]. Conversely, though, elevated IFNγ at tumour sites has been implicated in immune evasion[6]. Furthermore, elevated IFNγ signalling is observed in a significant proportion of melanoma and NSCLC lesions that progress on ICBT[7,8]. Moreover, tumours derived from cells chronically treated with IFNγ prior to implantation in syngeneic mice are resistant to ICBT[9]; while in vivo CRISPR screens revealed IFNγ signalling as a driver of ICBT resistance in multiple syngeneic mouse tumour implantation models[10]. The upregulation of major histocompatibility complex (MHC) and antigen-processing factors by the transcription factor Signal transducer and activator of transcription 1 (STAT1) downstream of IFNγ augments tumour antigenicity and thereby increases tumour cell recognition by T effector cells; in contrast, the duration and strength of anti-tumour responses are impeded by IFNγ-induced immunomodulatory molecules, including PD-L1, which confer immune homoeostasis[11]. In addition, induction of interferon regulatory factor 2 (IRF2), a STAT1 target gene product, in T cells also results in interferon-mediated T-cell exhaustion in multiple tumour types[12]. The identification of novel actionable targets mediating IFNγ-driven acquired resistance is urgently needed to improve the success of ICBT.

In this work, we investigate IFNγ-driven reprogramming of gene expression in tumour cells associated with acquired resistance to ICBT, therein demonstrating a role for the IFNγ target gene product poly-ADP ribosyl polymerase 14 (PARP14). Although less studied than other PARPs, PARP14 has recently emerged as a promising therapeutic target in chronic inflammation. As a STAT6 transcriptional co-activator, PARP14 polarises immune responses towards type 2 T helper ($T_H$2) mediated[13,14]; while in IFNγ-treated macrophages, pro-inflammatory differentiation is suppressed by PARP14 through inhibition of STAT1 phosphorylation and down-regulation of STAT1 target genes[15]. Although PARP14 is an established oncoprotein[16,17], its anti-inflammatory functions in the context of tumour immune evasion remain poorly characterised.

## Results

### Chronic IFNγ exposure drives resistance to α-PD-1 therapy and upregulates PARP14

Subcutaneous transplantation of mouse YUMM2.1 melanoma and CT26 and MC38 colon carcinoma cells into immunocompetent syngeneic mice gives rise to tumours that regress or stabilise to varying extents upon treatment with anti-PD-1 (α-PD-1) antibodies[18,19]. However, chronic exposure of tumour cells to IFNγ limits the effectiveness of immune checkpoint inhibitors such as α-PD-1, driving resistance to treatment[9]. To validate the role of chronic IFNγ exposure in α-PD-1 therapy resistance, we implanted syngeneic mouse hosts subcutaneously with either IFNγ-naïve, bovine serum albumin (BSA)-exposed YUMM2.1, CT26 and MC38 cells or with the same cells pre-treated continuously with 50 IU/mL IFNγ for at least 2 weeks. Once tumours were established (reached approximately 80 mm³), mice were subsequently treated with either α-PD-1 or IgG2a isotype control antibody (Fig. 1A). Chronic IFNγ pre-treatment did not affect tumour growth rates when mice were treated with control antibody (Fig. 1B–D). Tumours derived from all three cell lines without chronic IFNγ pre-treatment demonstrated delayed growth in response to α-PD-1 therapy for at least one-week post-treatment before mice eventually succumbed to progressive tumour growth, with tumours derived from YUMM2.1 responding best, followed by CT26 and then

MC38 (Fig. 1E–G). In contrast, IFNγ pre-treatment eliminated the ability of tumours to respond to α-PD-1 therapy significantly shortening survival (Fig. 1H–J and Supplementary Fig. 1A–D), indicating that adaptation to IFNγ promotes α-PD-1 therapy resistance.

Chronic IFNγ exposure induces constitutive (ligand-independent) target gene expression through epigenetic reprogramming[11,20,21]. To identify mechanisms of IFNγ-driven adaptive resistance to α-PD-1, we exposed human (A375 and 501-mel) and mouse (B16-F10, MC38, 5555, and YUMM2.1) tumour cell lines to IFNγ (20 IU/mL for human and 50 IU/mL for mouse cell lines) or BSA continuously for 2 weeks, and subsequently performed RNA sequencing (RNA-seq). In addition to well-established IFNγ target genes such as CD274, IRF1, and B2M, three members of the PARP family—PARP9, −12 and −14—were consistently upregulated in all cell models as well as in IFNG^high patient melanoma (comparing top 15% by IFNG expression to lowest 15% in the TCGA SKCM dataset) (Fig. 1K).

PARP14 was of interest to us as a possible targetable mediator of chronic IFNγ-driven resistance to α-PD-1, given that we have developed a potent and highly selective orally available small molecule inhibitor of PARP14 with effects on gene expression in tumour explants overlapping with those of α-PD-1[22]. In agreement with PARP14 being an IFNγ target gene, we observed PARP14 levels increasing in response to higher doses of IFNγ in multiple human and mouse tumour cell lines (Supplementary Fig. 1E). We also observed that chronic stimulation of these cell models with IFNγ resulted in sustained STAT1 expression and STAT1 activating phosphorylation coincident with augmented PARP14 expression compared to baseline or early (<24 h) IFNγ stimulation (Supplementary Fig. 1F). Indeed, addressing this in more detail in YUMM2.1 and CT26 cell models, we observed an inability of fresh IFNγ to further induce expression of protein markers of IFNγ stimulation in cells chronically exposed to IFNγ (Fig. 2A, B). Sequencing of mRNA (RNA-seq) extracted at the same time points combined with gene set enrichment analysis (GSEA)[23,24] also corroborated that IFNγ-driven gene expression changes were elevated above baseline by chronic stimulation reaching a plateau between 2-3-weeks, with no further stimulation by addition of fresh IFNγ possible beyond 2 weeks (Fig. 2C, D). Intriguingly, we also observed different signalling events enriched besides IFNγ signalling during chronic IFNγ treatment in both cell line models, including TGFbeta signalling, NF-kB in response to tumour necrosis factor (TNF) pathways, and inflammatory responses (Fig. 2C, D). Demonstrating the relevance of these IFNγ-adapted models to ICBT acquired resistance in patients, expression levels of sixteen genes commonly selectively upregulated in response to chronic IFNγ treatment (see methods section for how a chronic IFNγ signature was derived) associated with poor response to ICBT in a large cohort of patients with metastatic urothelial cancer who were treated with an anti-PD-L1 agent (atezolizumab)[25] (Supplementary Fig. 2A) and were typically associated with poor cancer patient survival (Supplementary Fig. 2B–G).

In keeping with PARP14 being a STAT1-regulated gene, in silico analysis indicated the location of multiple putative STAT1 binding sites on the PARP14 promoter (Supplementary Fig. 3A). Analysis of ChIP-seq data (retrieved from ENCODE project database)[26] confirmed the binding of STAT1 near the transcription start site of PARP14 in IFNγ-treated cells (Supplementary Fig. 3B). In A375 cells transfected with a reporter plasmid in which Gaussia luciferase is regulated by the PARP14 promoter, exposure to increasing IFNγ concentration enhanced luciferase activity. Furthermore, short interfering RNA (siRNA)-mediated STAT1 depletion in these cells impaired the ability of IFNγ to activate the reporter, confirming PARP14 induction through the IFNγ-STAT1 axis (Supplementary Fig. 3C). In addition, the mRNA abundance of PARP14 in both melanoma cells and patient samples (TCGA SKCM) positively correlated with that of STAT1, IFNG, and CD274 (Supplementary Fig. 3D–G).

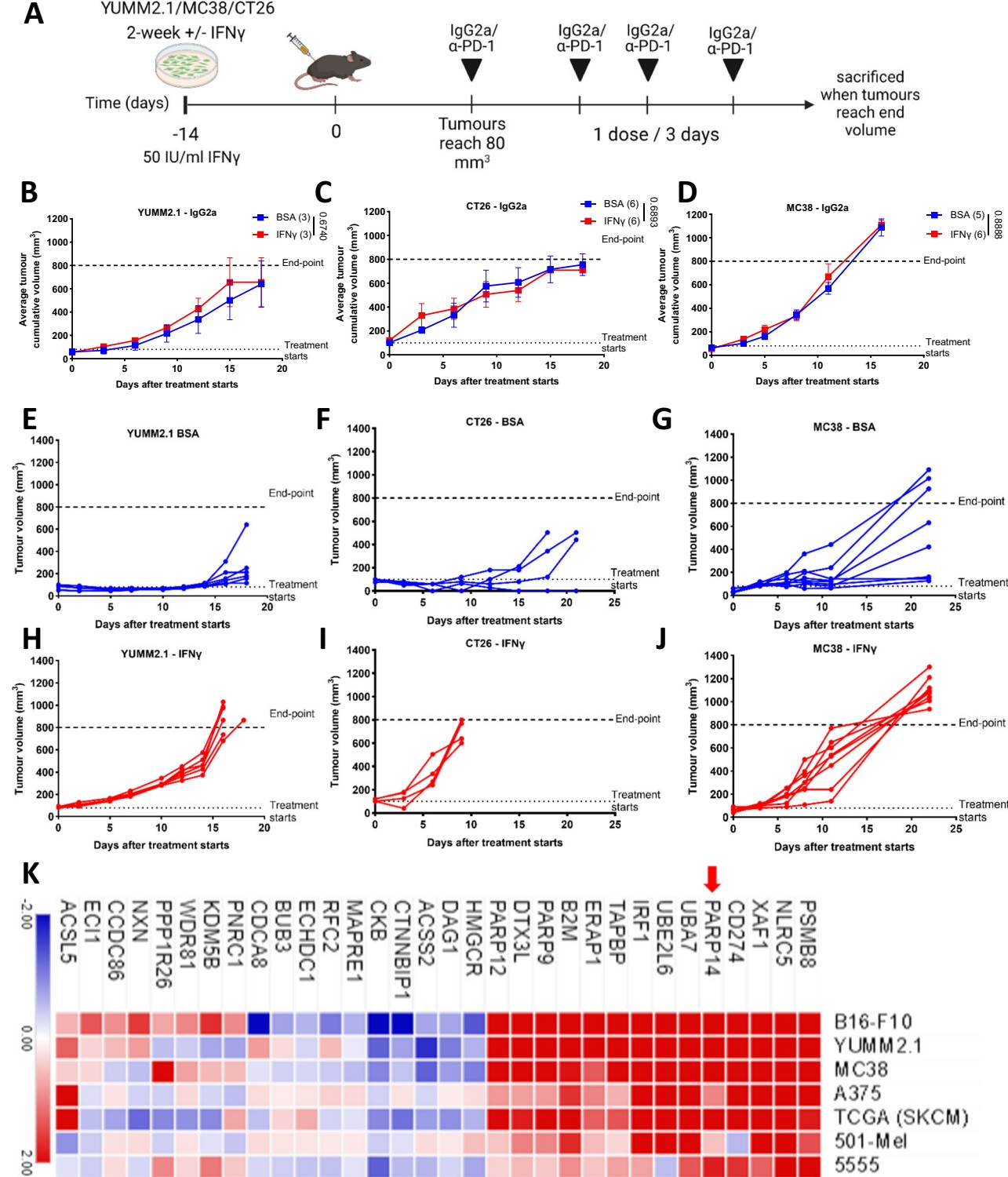

## PARP14 inhibition or depletion in tumour reverses adaptive resistance to α-PD-1 therapy

To address the role of PARP14 in chronic IFNγ-driven resistance to α-PD-1 and to demonstrate the potential of PARP14 as a therapeutic target capable of modulating α-PD-1 sensitivity, we treated mice implanted with IFNγ pre-treated YUMM2.1, CT26, or MC38 cells with α-PD-1 in combination with RBN012759, a highly selective PARP14 catalytic inhibitor[22] (PARP14i) (Fig. 3A). According to previous findings, twice daily dosing of mice with 500 mg/Kg RBN012759 achieves stable PARP14 suppression without adverse effects[22]. At this dose, PARP14i

strongly synergised with α-PD-1, with tumour regression and significantly extended survival observed in all three models (Fig. 3B and Supplementary Fig. 4A–E). A suppressive effect, albeit reduced, on YUMM2.1 tumour growth and extended mouse survival was observed using a lower dose of PARP14i in combination with α-PD-1 (Supplementary Fig. 4F). 25% of mice bearing YUMM2.1 tumours treated with a combination of α-PD-1 and 500 mg/Kg PARP14i exhibited durable tumour regression (up to 60 days post-treatment) (Fig. 3C). Additionally, at 2 months post cessation of combination therapy, all long-term survivors rejected the re-implantation of chronic IFNγ pre-treated

**Fig. 1 | Chronic IFNγ exposure drives resistance to α-PD-1 therapy and upregulates PARP14. A** YUMM2.1, CT26, and MC38 cells were implanted into 8–12-week-old wild-type syngeneic female mice after two-weeks pre-treatment with IFNγ (50 IU/mL) or BSA. Treatment with the control IgG2a or α-PD-1 antibody was initiated once tumour volume reached 80–100 mm³, with dosing every three days for a total of four doses. **B–D** Average tumour growth curve for **B** YUMM2.1 (BSA: $n = 3$; IFNγ: $n = 3$), **C** CT26 (BSA: $n = 6$; IFNγ: $n = 6$), and **D** MC38 (BSA: $n = 5$; IFNγ: $n = 6$) cells after initiating treatment with the control IgG2a antibody. Number of mice treated indicated in parentheses. The $p$-value for tumour growth was assessed for the last day of IgG2a treatment and were determined by an unpaired two-sided $t$-test with Welch's correction. The data were presented as the mean ± SEM. **E–G** The growth curve of each **E** YUMM2.1 ($n = 6$), **F** CT26 ($n = 4$), and **G** MC38 ($n = 8$) tumour pre-treated with BSA receiving α-PD-1 therapy. **H–J** The growth curve of each **H** YUMM2.1 ($n = 6$), **I** CT26 ($n = 4$), and **J** MC38 ($n = 8$) tumour pre-treated with IFNγ receiving α-PD-1 therapy. **K** Gene expression heatmap of differentially expressed genes (complete Euclidean HCL clustered; log2 fold change ≥ ±0.5; FDR ≤ 0.1) in mouse (B16-F10, YUMM2.1, MC38, 5555) or human (A375, 501-Mel,) tumour cell lines treated with chronic IFNγ (50 IU/mL for mouse and 20 IU/mL for human) compared to BSA treatment. Three independent cell line samples were sequenced for both conditions and the average for each cell line is shown. Heatmap also includes differential gene expression comparing melanoma patient samples with the 15% highest IFNG expression level with the 15% lowest (data retrieved from TCGA SKCM RNA sequencing data using Broad GDAC Firehose). Source data are provided as a Source Data file.

---

YUMM2.1 cells (Fig. 3D), indicating the induction of anti-tumour immune memory.

Next, we addressed the extent to which CD8 + T cells control tumour growth in α-PD-1/PARP14i combination-treated animals (Fig. 3E, F). We found that significantly depleting CD8+ cells (confirmed by flow cytometry) through systemic administration of α-CD8 antibody permitted the progression of combination therapy-treated tumours (Fig. 3G, H). To determine whether PARP14 inhibition directly affects T cells, we isolated T cells from BALB/c mouse spleens and stimulated them using CD3 + CD28 antibodies in the presence of DMSO or RBN012759. Effects of PARP14 inhibition were examined at two time points: the first following the period of initial stimulation (48 h), and the second after a rest period of 7 days and restimulation for 14 h or 96 h (Fig. 4A). Acute stimulation of T cells in the presence of PARP14i significantly increased expression of *Parp14* mRNA, but did not significantly increase *Stat1*, *Irf1*, or *Cd274* mRNA compared with the DMSO control (Fig. 4B–E), although a tendency for their increase was noted. Because PARP14 has been shown to have epigenetic effects through modification of HDAC activity[27] and epigenetic effects often require repeat stimulation to reveal the induced changes in gene accessibility, we restimulated these T cells after they came to rest. After rest and restimulation for 14 h with CD3 + CD28 in the absence of additional PARP14i, cytokine production was assessed by flow cytometry. Significant increases in the percentages of IFNγ and TNFalpha (TNFα) producing CD4 and CD8 T cells were seen (Fig. 4F, G; Supplementary Fig. 5 for gating strategy). Conversely, significant decreases in latency associated peptide (LAP)+ (latent transforming growth factor-beta (TGFβ)), and interleukin 10 (IL-10) + CD4 and CD8 T cells were also seen (Fig. 4H, I). Interestingly, PARP14i-pre-treated CD8 + T cells appeared to be more proliferative expressing a higher percentage of Ki-67+ (Fig. 4J; Supplementary Fig. 6 for gating strategy). We concluded that PARP14i treatment could induce a pro-inflammatory phenotype in both CD4+ and CD8 + T cells. In addition, PARP14i reduced the production of suppressive factors such as TGFβ and IL-10, possibly contributing to the reduction in regulatory T (Treg) cells we observed in PARP14i-treated tumours.

To address the contribution of PARP14 expression in tumour cells to mediating α-PD-1 resistance, we expressed short-hairpin (sh) RNA targeting PARP14 (shPARP14) for RNA interference-mediated downregulation in YUMM2.1 and MC38 cells. We then implanted IFNγ pre-treated shNTC- or shPARP14-expressing cells into mice and applied the same IgG2a or α-PD-1 treatment regimen described above (Fig. 5A). PARP14 depletion in these two cell models had no significant effect on tumour formation or tumour growth potential in control IgG2a-treated mice (Supplementary Fig. 7A, B). However, PARP14 depletion restored responsiveness to α-PD-1 therapy (Fig. 5B–E and Supplementary Fig. 7C). Quantitation of *Parp14* mRNA expression in bulk-tumour by RT-PCR analysis revealed that while expression was still significantly lower in endpoint YUMM2.1 tumours expressing shPARP14 compared to tumours expressing shNTC (Supplementary Fig. 7D), this was not the case for MC38 tumours (Supplementary Fig. 7D), suggesting a selection for elevated *Parp14* mRNA expression or a loss of the shRNA in MC38 tumours treated with α-PD-1 and perhaps accounting for the less robust effect of PARP14 depletion in this model.

## Chronic IFNγ exposure reshapes the tumour immune infiltrate through PARP14

Given that IFNγ stimulation can lower tumour-associated antigen expression through stimulating both immune proteasome degradation and dedifferentiation with a resultant decrease in infiltration by activated CD8+ effector T cells[7]; suppress NK and CD8 + CTL activation through increasing expression of inhibitory non-cognate MHC-I molecules such as human leucocyte antigens class I histocompatibility antigen, alpha chain E (HLA-E) (equivalent to Qa-1b/H2-T23 in mice)[10]; and exert immunosuppressive effects through increasing expression of immune checkpoint ligands, Indoleamine 2, 3-dioxygenase 1 (IDO1) and chemokine (C-C motif) ligand (CCL8) in tumours[28,29], we reasoned that chronic IFNγ stimulation might alter the tumour immune microenvironment in our syngeneic tumour models. To assess this, in the first instance, we investigated gene expression differences in tumours from IFNγ-naïve and chronic IFNγ-treated YUMM2.1 cells (treated with IgG2a control antibodies) by undertaking RNA-seq and GSEA. This revealed that the IFNγ-treated cells down-regulated several inflammatory signalling pathways (Supplementary Fig. 8A). Moreover, we also performed computational immunophenotyping using ImmuCellAI cell-type enrichment analysis[30], which indicated that immune score, T cells, CD8 T cells, and CD8 central memory T cells (Tcm), were down-regulated while Granulocytes and myeloid dendritic cells (MoDC) were upregulated in tumours pre-treated with IFNγ (Supplementary Fig. 8B–G).

To corroborate the above in silico immunophenotyping, we next profiled the immune infiltrate of subcutaneous tumours derived from IFNγ-naïve YUMM2.1 cells expressing shNTC or chronic IFNγ pre-treated YUMM2.1 cells expressing either shNTC or shPARP14 (Fig. 5F) by flow cytometry using fluorescent labelling for a panel of T cell markers including TCRαβ, TCRγδ, CD45, CD25, and FoxP3 (Supplementary Fig. 9A for gating strategy). Tumours derived from chronic IFNγ-pre-treated YUMM2.1 cells exhibited a significantly lower percentage of T cells (TCRαβ⁺) and a higher percentage of regulatory T (Treg) cells (Foxp3⁺, CD25high) relative to tumours derived from either IFNγ-naïve cells but also to shPARP14-expressing IFNγ-pre-treated YUMM2.1 cells (Fig. 5G–J), implying that PARP14 might contribute to the immunosuppressive tumour microenvironment induced by chronic IFNγ pre-treatment. Interestingly, NK cells were upregulated in tumours derived from chronic IFNγ-pre-treated YUMM2.1 cells, again reversed upon PARP14 knockdown, but there was no dysregulation in gamma-delta T cells (Supplementary Fig. 9B, C).

Next, we assessed whether PARP14i alone or in combination with α-PD-1 could reverse the immunosuppressive effects of chronic IFNγ pre-treatment (Fig. 6A). Compared to control-treated tumours, the combination of α-PD-1 and PARP14i elicited an increased percentage of CD8⁺ T cells and a decreased percentage of Treg cells, leading to a significant increase in the ratio of CD8⁺ Granzyme B⁺ (GzmB⁺) cytotoxic lymphocytes (CTLs) to Treg cells (Fig. 6B–E and

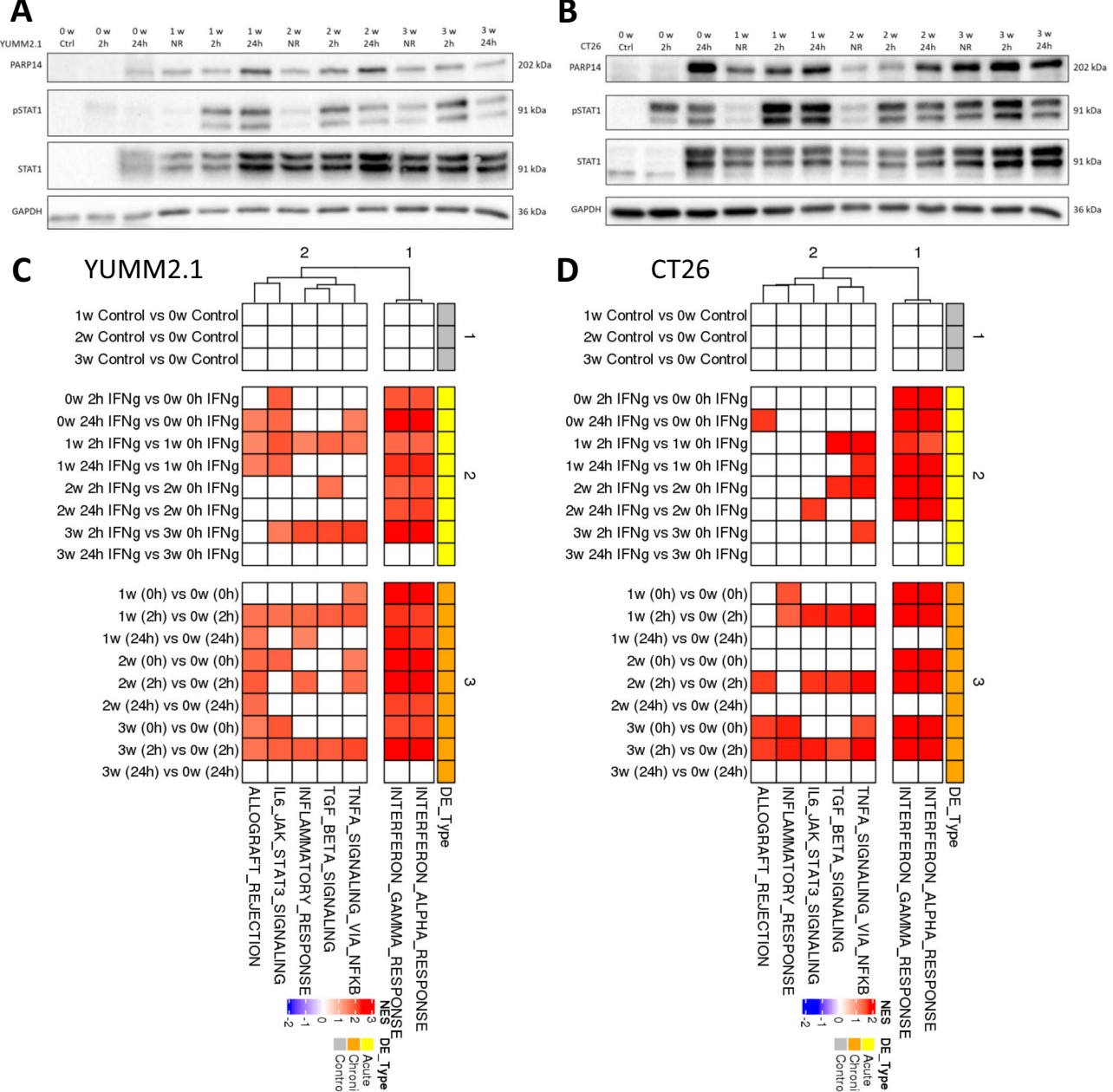

**Fig. 2 | PARP14 expression reaches a plateau after 3-weeks IFNγ treatment and multiple inflammatory-related pathways are significantly upregulated after chronic IFNγ treatment. A** YUMM2.1 and **B** CT26 tumour cells were treated continuously with IFNγ for 3-weeks being periodically restimulated as indicated. PARP14, pSTAT1 and STAT1 protein expression were determined via western blot with GAPDH used as a loading reference. Samples from left to right: 0 w Ctrl (*n* = 3): 0-week no treatment; 0 w 2 h (*n* = 3): IFNγ treatment for 2 h (2 h) at 0-week; 0 w 24 h (*n* = 3): IFNγ treatment for 24 h (24 h) at 0-week; 1 w NR (*n* = 3): IFNγ treatment for 1-week with no restimulation (NR); 1 w 2 h (*n* = 3): IFNγ treatment for 1-week plus restimulation of IFNγ for 2 h; 1 w 24 h (*n* = 3): IFNγ treatment for 1-week plus restimulation of IFNγ for 24 h; 2 w NR (*n* = 3): IFNγ treatment for 2-week with NR; 2 w 2 h (*n* = 3): IFNγ treatment for 2-week plus restimulation of IFNγ for 2 h; 1 w 24 h (*n* = 3): IFNγ treatment for 2-week plus restimulation of IFNγ for 24 h; 3 w NR (*n* = 3): IFNγ treatment for 3-week plus restimulation of IFNγ for 2 h; 3 w 24 h (*n* = 3): IFNγ treatment for 3-week plus restimulation of IFNγ for 24 h. The images were representatives of 1 of 3 independent experiments. GSEA of RNA-seq data from **C** YUMM2.1 and **D** CT26 cell lines treated in triplicate as in (**A**) and (**B**), highlighting different hallmark processes enriched at different time points. Source data are provided as a Source Data file.

Supplementary Fig. 10 for gating strategy). Moreover, compared to control-treated tumours, the combination of α-PD-1 and PARP14i demonstrated a significant increase of CD4+ and CD8 + T cells expressing surface inhibitory receptors PD-1, TIM-3, and LAG-3 (Fig. 6F–I). Collectively, the above findings show that PARP14 antagonism potentiates the immunostimulatory effect of α-PD-1 in an otherwise immunosuppressive tumour microenvironment established by chronic IFNγ-signalling.

We also contrasted gene expression in tumours derived from chronic IFNγ stimulated YUMM2.1 cells treated with α-PD-1 monotherapy with α-PD-1 + PARP14i combination therapy treated by sequencing mRNA from bulk tumours. GSEA revealed that the combination therapy upregulated numerous inflammatory signalling pathways (Fig. 7A). Furthermore, Ingenuity Pathway Analysis indicated that *STAT1, IFNG*, and *TNF* responses were strongly activated when PARP14 was also inhibited (Fig. 7B). Additionally, leucocyte migration

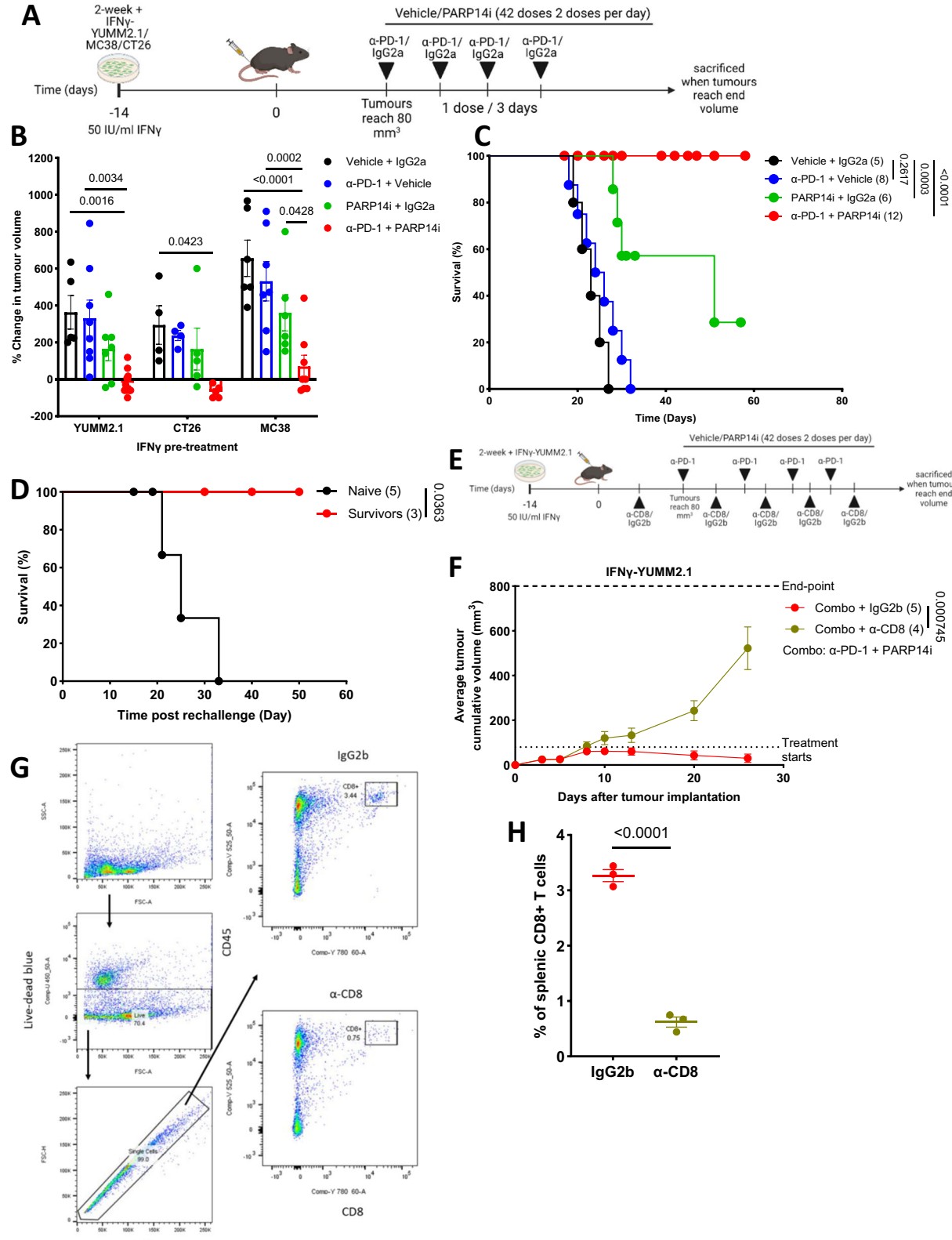

and activation of antigen-presenting cells were strongly activated and tumourigenesis-related processes strongly down-regulated following PARP14i treatment (Fig. 7C). Furthermore, computational immunophenotyping indicated that tumours undergoing combination therapy and PD-1 monotherapy also demonstrated a significant upregulation of infiltration score, T cells, M1 macrophage, CD8 Tcm, CD8 effector memory T cells (Tem), CD8 exhausted T cells (Tex), and down-regulation of granulocytes compared with controls; interestingly, only combination therapy had a significant upregulation of MoDCs and CD8 cytotoxic T cells (TC), comparing with the control and α-PD-1 monotherapy respectively (Fig. 7D–L), consistent with PARP14i contributing to increased immune infiltration.

**Fig. 3 | PARP14 pharmacological antagonism reverses adaptive resistance to α-PD-1 therapy. A** Chronic IFNγ pre-treated YUMM2.1/MC38/CT26 cells were subcutaneously implanted into 8–12-week-old wild-type syngeneic female mice. Treatment with either α-PD-1 or IgG2a antibody was initiated once tumour volume reached 80–100 mm³, with antibodies administered every three days for a total of four doses. In parallel, the animals also received two daily doses of the PARP14 inhibitor (PARP14i) RBN012759 or vehicle for a total of three weeks. **B** The percentage change in tumour volume between the first dose of treatment and the administration of the final α-PD-1 dose of mice receiving implants of chronic IFNγ pre-treated YUMM2.1 (Vehicle + IgG2a: n = 5; α-PD-1 + Vehicle: n = 8; PARP14i + IgG2a: n = 6; α-PD-1 + PARP14i: n = 12), CT26 (Vehicle + IgG2a: n = 4; α-PD-1 + Vehicle: n = 4; PARP14i + IgG2a: n = 5; α-PD-1 + PARP14i: n = 5), and MC38 (Vehicle + IgG2a: n = 6; α-PD-1 + Vehicle: n = 7; PARP14i + IgG2a: n = 6; α-PD-1 + PARP14i: n = 8). The adjusted p-values were determined by two-way ANOVA Tukey's test and the data were presented as the mean ± SEM. **C** Kaplan–Meier plots of IFNγ pre-treated YUMM2.1 (Vehicle + IgG2a: n = 5; α-PD-1 + Vehicle: n = 8; PARP14i + IgG2a: n = 6; α-PD-1 + PARP14i: n = 12) in different treatment groups with the number of mice in each arm indicated in parentheses and the p-values were determined by Log-rank (Mantel-Cox) test. **D** Survivors of α-PD-1 and PARP14i combinatorial therapy were maintained for 60 days and then re-implanted with IFNγ pre-treated YUMM2.1 cells (Naïve: n = 5; Survivors: n = 3). Age-matched naive mice act as a control group. Survival is shown for both groups. The p-value was assessed by Log-rank (Mantel-Cox) test. **E** Tumour-bearing mice received a total of four doses of α-PD-1 antibodies, 42 doses of PARP14i (twice daily for three weeks), and five doses of α-CD8 or IgG2b antibodies (a single dose every five days). **F** Average cumulative tumour volume over the course of treatment (Combo + IgG2b: n = 5; Combo + α-CD8: n = 4). The p-value was assessed at day 24 post-tumour implantation by two-sided unpaired t-test and the data were presented as the mean ± SEM. **G** Flow cytometry gating strategy for assessing the efficiency of splenic CD8 + T cell depletion by α-CD8. **H** Frequency of CD8+ cells among splenic T cells for animals receiving combination therapy with IgG2a (green: n = 3) or α-CD8 (red: n = 3). The p-value was assessed by two-sided unpaired t-test and the data were presented as the mean ± SEM. Source data are provided as a Source Data file.

## PARP14 is a negative feedback regulator of IFNγ signalling

PARP14 down-regulates STAT1 in IFNγ-stimulated macrophages and thereby antagonises IFNγ-induced macrophage polarisation[15]. Using immunoprecipitation, we detected an interaction between STAT1 and PARP14 in three melanoma cell lines—MV3, LOX-IMVI, and YUMM2.1 (Supplementary Fig. 11A–C). We hypothesised that PARP14 might also act, therefore, as a negative feedback regulator in tumour cells, thereby antagonising IFNγ-stimulated tumour cell immunogenicity. In keeping with this hypothesis, we found that phospho-STAT1 (pSTAT1), STAT1, and STAT1 target gene products PD-L1, MHCI, TAP1, and TAP2 were enriched in shPARP14-expressing, chronic IFNγ-treated YUMM2.1 and MC38 cells compared to shNTC-expressing cells (Supplementary Fig. 11D). Similarly, pharmacological antagonism of PARP14 using either RBN012759 or the proteolysis targeting chimera (PROTAC) PARP14 inhibitor RBN012811[22,31] at nanomolar concentrations in chronic IFNγ-treated A375, 501-Mel, YUMM2.1, or MC38 cells resulted in elevated levels of pSTAT1 and STAT1 target gene products with only minor perturbations of the growth of these cell lines (Fig. 8A–C). Intriguingly, while the expected depletion of PARP14 protein occurred following the degradation-inducing RBN012811 treatment, application of the catalytic inhibitor RBN012759 led to elevated levels of PARP14 protein, consistent with PARP14 being itself a STAT1 activated target (Fig. 8A–C). RBN012759 treatment did not interfere with STAT1–PARP14 interaction (Supplementary Fig. 11A–C). Moreover, RNA-seq and subsequent GSEA revealed that PARP14 inhibition enhanced inflammatory signalling (Fig. 8D). Quantitative PCR (qPCR) confirmed a significant increase of mRNA expression for the chemokine ligands Cxcl10 and Cxcl11 (Fig. 8E), supporting our hypothesis that PARP14 inhibition enhances IFNγ signalling in tumour cells and upregulates immune cell infiltration into tumours.

## PARP14 levels are augmented in tumours spontaneously relapsing after α-PD-1 treatment wherein it mediates resistance

To address the role of PARP14 in spontaneously arising acquired resistance to α-PD-1 therapy, we firstly validated whether PARP14 expression could be induced by α-PD-1 therapy in our syngeneic mice models, as expression data from human melanoma biopsies[32] indicated a modest but significant increase in PARP14 mRNA in α-PD-1 on-treatment melanoma biopsies compared to pre-treatment biopsies that correlated with increased IFNG and STAT1 mRNA (Supplementary Fig. 12A, B). Following establishment of tumours derived from IFNγ-naïve YUMM2.1 and MC38 cells, mice were treated with two doses (spaced three days apart) of IgG2a or α-PD-1 antibodies. Tumours were harvested within 24 h of the last dose of treatment. qPCR did not reveal a significant upregulation of Parp14 in bulk-tumour mRNA from α-PD-1 on-treatment tumours compared to control tumours (Supplementary Fig. 12C, D). However, when we grouped YUMM2.1 and MC38 tumour specimens by the median level of Ifng mRNA (Ifng^high versus Ifng^low) regardless of treatment, Parp14 expression was significantly higher in Ifng^high tumours. We made similar observations for Stat1 and other STAT1 target genes including Irf1, Cxcl10, and Cxcl11 (Supplementary Fig. 12E, F), which suggested that PARP14 was induced specifically in IFNγ-inflamed tumours but independently of α-PD-1. Consistent with α-PD-1 not inducing PARP14 in responding tumours, PARP14i treatment failed to inhibit growth of tumours derived from IFNγ-naïve YUMM2.1 cells nor enhanced the activity of α-PD-1 when both drugs were co-administered (Supplementary Fig. 13A–C).

Despite PARP14 induction in tumours derived from IFNγ-naïve YUMM2.1 cells not depending on α-PD-1 treatment, we found by qPCR analysis that Ifng, Stat1 and other IFNγ target genes, including Parp14, were significantly increased in tumours that regrew following α-PD-1 treatment compared to control tumours (Fig. 9A). Comparable gene expression changes were observed in short-term melanoma cell cultures derived from ICBT-progressing patient lesions with elevated IFNγ-signalling compared to cultures where intrinsic IFN-signalling was minimal[7] (Fig. 9B). Following bulk-tumour mRNA sequencing, GSEA revealed that the α-PD-1-relapsing tumours upregulated immune-related inflammatory signalling pathways, regardless of comparing with control or α-PD-1-responding conditions. In particular, both IFNγ and IFN-α signalling pathways were upregulated significantly in the α-PD-1-relapsing tumour (Fig. 9C). This suggested that α-PD-1 relapsing tumours might have an enriched immune infiltration status yet failed to control tumour growth. Indeed, computational immunophenotyping revealed that, compared with the control, the α-PD-1-relapsing tumours showed a higher infiltration score and T cell, CD8 T cell, CD8 Tcm, CD8 Tem, and CD8 Tex scores (Fig. 9D–I). In addition, whereas the α-PD-1-responding tumours had a variable level of different immune subsets scores, relapsing tumours had a significantly lower score of macrophage and naïve CD8 T cell, which might suggest that α-PD-1 responding tumours had fewer differentiated or functional innate and adaptive immune cells in the tumour microenvironment (Fig. 9J, K). This correlated with higher expression of immunosuppressive molecules in the α-PD-1-relapsing condition (Fig. 9L). Moreover, PARP14 expression in short-term melanoma cell cultures derived from ICBT-progressing lesions was significantly negatively correlated with the infiltration of B cells and NK cells in the tumour microenvironment, suggesting that the higher PARP14 expression in tumour cells, the colder the tumour immune microenvironment (Supplementary Fig. 13D, E).

Given that IFNγ signalling, including Parp14 expression, was upregulated in tumours relapsing following α-PD-1 treatment, we next addressed whether these tumours were sensitive to Ruxolitinib (a JAK inhibitor) administered at a dose designed to suppress only tumour

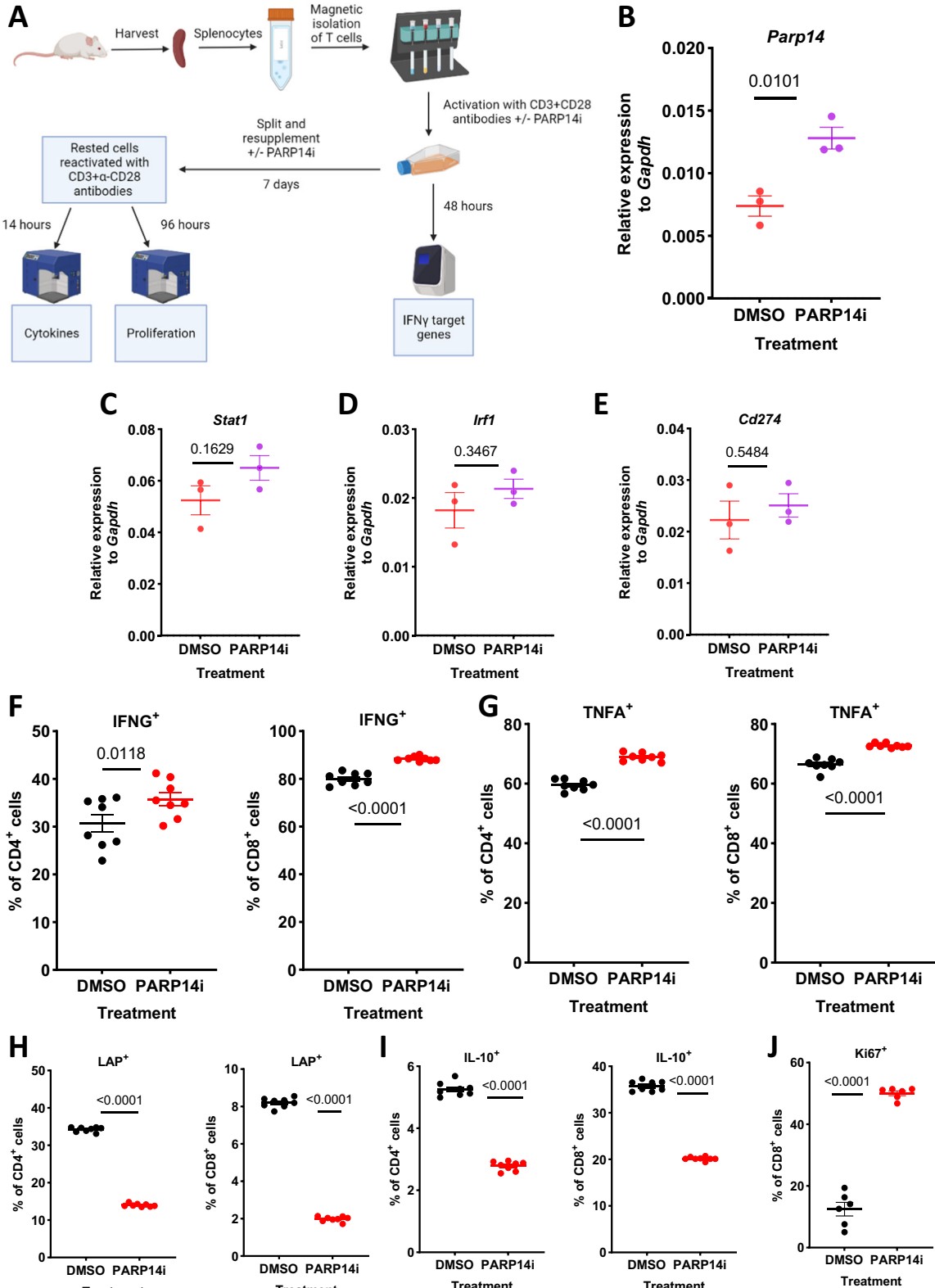

cell intrinsic IFNγ signalling[9] as well as to RBN012759. IFNγ-naïve YUMM2.1 cells were implanted subcutaneously, and subsequently tumour-bearing mice received two doses of α-PD-1 antibodies three days apart. Following the final dose of α-PD-1, tumours were permitted to regrow to their pre-treatment size (regrowing tumours are insensitive to further doses of α-PD-1; Supplementary Fig. 14A, B) and randomised onto one-week control vehicle and IgG antibodies, PARP14i

(14 doses), α-PD-1 (a further 2 doses) and PARP14i, and JAKi (7 doses) treatment (Fig. 10A). All treatments suppressed tumour growth and extended survival significantly (Fig. 10B–D). Moreover, there was no significant difference between treatment types. Interestingly, the combination therapy of α-PD-1 and PARP14i did not synergise as they did in the IFNγ pre-treatment model (Fig. 10B–D). The growth of tumours derived from IFNγ-naïve CT26 cells that regrew following α-

**Fig. 4 | PARP14 inhibition promotes pro-inflammatory CD4+ and CD8 + T cell phenotype. A** BALB/C mice were sacrificed and their spleens harvested. Isolated T cells were firstly stimulated with α-CD3 and α-CD28 antibodies, with 1 µM RBN012759 or DMSO. After 48 h, sample aliquots were processed for RT-qPCR analyses, while remaining cells were passaged refreshing the DMSO/ PARP14i for an additional 7 days. Subsequently, cells were re-activated with α-CD3 and α-CD28 antibodies and stained for cytokines after 14 h and for proliferation marker Ki-67 after 96 h. **B–E, B** *Parp14*, **C** *Stat1*, **D** *Irf1*, **E** *Cd274* mRNA expression in cells treated with DMSO (n = 3) or PARP14i (n = 3) relative to the housekeeping gene *Gapdh*. The data were presented as mean ± S.E.M. and the *p*-values were determined by two-sided unpaired *t*-test. **F–I** Percentage of DMSO (n = 8) and PARP14i (n = 8) pre-treated cells that were **F** CD4 + IFNG+ (left) and CD8 + IFNG+ (right), **G** CD4 + TNFA+ (left) and CD8 + TNFA+ (right), **H** CD4 + LAP+ (left) and CD8 + LAP+ (right), **I** CD4 + IL-10+ (left) and CD8 + IL-10+ (right). The data were presented as mean ± S.E.M. and the *p*-values were assessed by two-sided unpaired *t*-test. **J** Percentage of DMSO (n = 6) and PARP14i (n = 6) pre-treated cells that were CD8+ Ki- 67+ cells. The data was presented as mean ± S.E.M. and the *p*-value was assessed by two-sided unpaired *t*-test. Source data are provided as a Source Data file.

PD-1 administration was also suppressed by PARP14i treatment (Supplementary Fig. 14C–E).

## Discussion

The contribution of IFNγ signalling to ICBT resistance remains controversial[33]. The induction of IFNγ and its target genes are sensitive and robust prognostic markers of ICBT response[5,34]. Contrasting *IFNG* and *STAT1* mRNA abundance in pre- versus on-treatment melanoma biopsies[32] also supported that IFNγ-signalling is induced by ICBT. As such, insensitivity to IFNγ might be predicted to benefit tumour cells in the context of ICBT. In keeping with this, Gao and colleagues showed that tumours from patients resistant to α-CTLA-4 therapy harboured genomic defects in IFNγ pathway components, including copy-number loss of IFNγ pathway genes (e.g., *IFNGR1/2*, *IRF1*, and *JAK2*) and amplification of IFNγ pathway inhibitors (e.g., *SOCS1* and *PIAS4*)[35], while whole-exome sequencing of patient biopsies revealed loss-of-function mutations in Janus kinase 1 (JAK1) and JAK2 in patients with primary and acquired resistance to PD-1 blockade[36,37]. Although these studies suggest that inactivation of IFNγ pathway components is a major cause of ICBT resistance, this conclusion was based on limited patient samples, and subsequent studies failed to detect these changes at a significant frequency in larger populations[32,38,39]. In contrast, metanalysis by Song and colleagues confirmed that loss of IFNγ signalling in tumour cells made tumours more susceptible to the host immune system in mouse syngeneic models despite causing resistance to immune effector cells in vitro, and revealed that patient tumours developing mutations in IFNγ signalling before treatment were more likely to respond to ICBT[40]. Furthermore, the vast majority of primary melanoma cell cultures from ICBT-progressing primary lesions responded robustly to IFNγ[7], as did all the cell lines used in our study, suggesting that IFNγ signalling is preserved despite immunoediting. Further, more abundant IFNγ target gene products in plasma from patients undergoing ICBT predicted relapse and poorer survival[10]. While IFNG.GS gene signature in T cells could be used to predict the prognosis of cancer patients treated with ICBT[9]; interestingly, we found that genes consistently selectively upregulated during chronic IFNγ exposure in our YUMM2.1 and CT26 tumour models predicted shorter survival of metastatic urothelial cancer patients treated with α-PD-L1 therapy. Our findings and those from other groups clearly show, therefore, that chronic IFNγ signalling causes resistance to ICBT, and suppressing these effects could have great therapeutic application.

Currently, three directions are being advocated to address IFNγ-driven resistance to ICBT[33]: (1) augmenting tumour antigen presentation, (2) inhibiting the action of T cell inhibitory receptors (TCIRs), or (3) increasing TNF signalling in tumours. However, each direction has its own advantages and shortcomings in improving the clinical response of cancer patients. Administering IFN-agonists could enhance the antigen presentation machinery components expression of tumour cells, thereby strengthening T-cell neoantigen recognition and overall anti-tumour response. However, it could also reduce NK-cell-mediated cytotoxicity. Administering JAKi could eliminate TCIRs, but it may also worsen patients' health due to myelo- and immunosuppression and NK-cell-mediated cytotoxicity[41,42]. Administering therapeutic agents that target

tumour-intrinsic cell death signalling could lower the cytotoxic threshold of tumour cells to TNF and IFNγ, increasing their susceptibility to immune attack. However, some studies raise concerns about IFNγ or TNF systematic upregulation promoting tissue damage or other autoinflammatory syndromes in patients[43,44]. Other inhibitors of IFNγ signalling and mediators of IFNγ-dependent immunoregulation exist that may be appropriate drug targets, although appropriate drugs have yet to be developed. Thus, depletion of the JAK-STAT signalling regulator LNK impaired tumour growth and potentiated α-PD-1 responses by relieving LNK-mediated STAT1 inhibition[45]. Similarly, inhibition of receptor-interacting serine threonine kinase 1 (RIPK1) enhanced STAT1 signalling and the activation of cytotoxic T cells contributing to anti-tumour activity[46]. Chronic IFNγ signalling consistently induced expression of Qa-1b/HLA-E, a ligand for the cytotoxic lymphocyte inhibitory receptor NKG2a/CD94, in mouse tumour models, thereby conferring resistance to α-PD-1 therapy[10]. In this study, we discovered that PARP14i could be a potential pharmacological candidate for addressing IFNγ-driven resistance to ICBT.

RNA-seq analysis revealed upregulation of *PARP14* expression in multiple human and mouse tumour cell lines chronically exposed to IFNγ as well as in *IFNG*high melanoma tumours, suggesting that *PARP14* is an IFNγ target gene. Indeed, data we present imply that *PARP14* is a direct STAT1 target. Consequently, induction of *PARP14* mRNA mirrored *IFNG* and *STAT1* mRNA in on-treatment melanoma biopsies. However, *PARP14* mRNA levels could not predict the depth or duration of clinical response among responders to ICBT in the Riaz et al. cohort[32]. This may reflect several confounding factors, including a limited number of samples available for analysis; highly heterogeneous tumours in which only small regions were sampled; sampling of different lesions pre- and on-treatment; and finally, biopsies that were not necessarily sampled at the point of relapse, which is when our in vivo data indicate that *PARP14* induction may be at its highest and exerting its greatest influence. Indeed, expression data from melanoma progressing lesions reveals a significant fraction of lesions progressing on ICBT wherein tumour cell intrinsic IFNγ signalling and therein PARP14 expression is elevated[7].

Our findings provide insight into possible clinical contexts wherein PARP14 antagonism might be at its most effective. PARP14 inhibitor treatment would not be efficacious as a monotherapy, as PARP14 antagonism or knockdown did not affect the ex vivo growth of the tumour cell lines we evaluated, nor did monotherapy significantly alter their tumour growth potential, even following chronic IFNγ stimulation. As such, PARP14 activity is redundant to other factors, notably to PD-1 signalling, in mediating immunosuppression. Combining PARP14i with α-PD-1 treatment also did not seem to alter the course of response unless tumour cells had already been driven to a resistant state by chronic exposure to IFNγ pre-implantation or were progressing upon α-PD-1 treatment, a condition associated with high tumour-intrinsic IFNγ signalling. Based on these observations, it is recommended to prioritise PARP14i for clinical trials in cancer patients progressing on ICBT, especially when increased IFNγ signalling is detected in the tumour or plasma. Moreover, PARP14i treatment was more successful than *PARP14* knockdown in tumour cells alone in

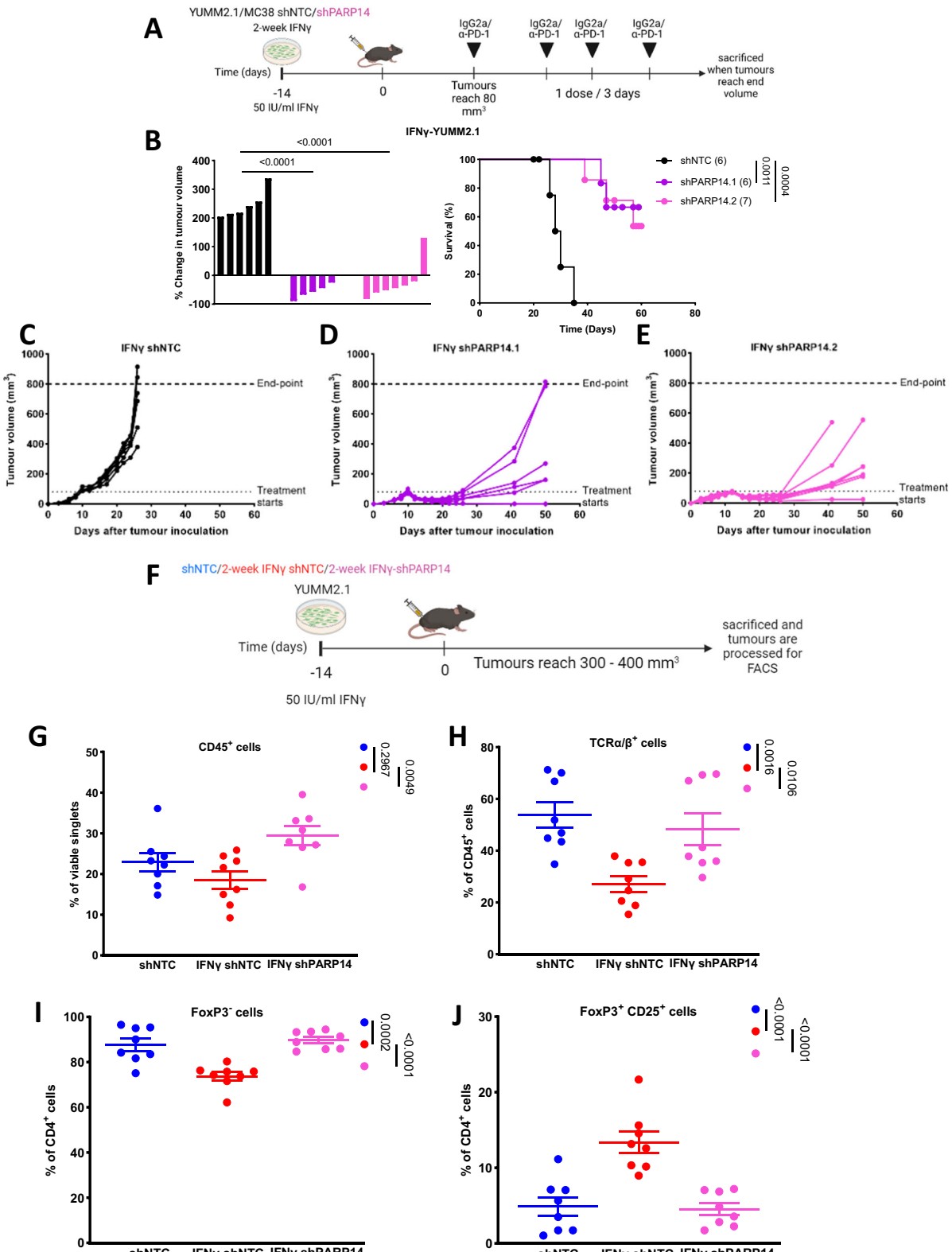

restoring α-PD-1 sensitivity. This could reflect a more potent suppression achieved by the inhibitor compared to knockdown but may also reflect the combined effect of antagonising PARP14 in tumour and host cells, as PARP14 is expressed in multiple immune cells and other normal cell types[13,14,47]. Indeed, we observed a more inflammatory phenotype in T cells stimulated following extended treatment with PARP14i. Moreover, previous findings have demonstrated that PARP14

might perform cancer-promoting functions in stromal cells present in the TME[14,15,22].

We explored the effects of IFNγ preconditioning and PARP14 depletion on the composition and activation status of the tumour immune infiltrate. We found that chronic IFNγ treatment of YUMM2.1 cells drastically remodelled the TME. Specifically, these tumours were infiltrated by significantly fewer CD45+ immune cells,

**Fig. 5 | PARP14 depletion in tumour cells reverses adaptive resistance to α-PD-1 therapy while reversing chronic IFNγ driven immune regulatory effects.** **A** Chronic IFNγ pre-treated YUMM2.1 cells expressing two independent PARP14-targeting shRNAs (shPARP14) (shPARP14.1: $n = 6$; shPARP14.2: $n = 7$) or a non-target control shRNA (shNTC) (shNTC: $n = 6$) were subcutaneously implanted into 8–12 weeks-old wild-type C57BL/6 female mice. Treatment of tumour-bearing mice was initiated once tumour volume reached ~80 mm³, with dosing every three days for a total of four doses. **B** The percentage tumour volume change between the first α-PD-1 treatment dose and the final dose (left) and Kaplan–Meier plots for these mice (right). The p-values were assessed by (left) the one-way ANOVA Dunnett's test and

(right) the Log-rank (Mantel-Cox) test. **C–E** Individual tumour growth rates for **C** shNTC, **D** shPARP14.1, and **E** shPARP14.2 expressing cells. **F** 8–12-week-old wild-type C57BL/6 mice were subcutaneously implanted with IFNγ-naïve YUMM2.1 cells expressing shNTC ($n = 8$) or chronic IFNγ pre-treated YUMM2.1 cells expressing shNTC ($n = 8$) or shPARP14 ($n = 8$). Tumours were allowed to grow to 300–400 mm³ and then dissected and disaggregated for analysis by flow cytometry. **G–J** Populations of **G** total immune cells (CD45⁺), **H** T cells (TCRαβ⁺), **I** CD4 effector T cells (CD4⁺ FoxP3⁻), and **J** regulatory T cells (Tʳᵉᵍ cells; CD25⁺FoxP3⁺) in the tumour infiltrate. The data were presented as mean ± S.E.M and the p-values were assessed by one-way ANOVA Dunnett's test. Source data are provided as a Source Data file.

including TCRαβ⁺ cells, whereas Treg cells were proportionately enriched, mirroring the ability of lymph node metastases that are likewise exposed to chronic IFNγ stimulation to induce or promote the recruitment of Treg cells[48]. Importantly, PARP14 silencing in IFNγ pre-treated tumours reversed these alterations in the TME. These effects of silencing PARP14 on the TME were consistent with restoring sensitivity to α-PD-1, as pre-existing immune infiltration and high expression levels of immune-related genes and low numbers of T$_{reg}$ cells predict good response to α-PD-1 therapy in patients[49–52]. Subsequently, we determined that the synergy between α-PD-1 therapy and PARP14i may also be explained by effects on the composition of the tumour immune infiltrate. Using the IFNγ pre-treated YUMM2.1 model, we found that the combination therapy led to an increase in intratumoral CD45⁺ immune cells and CD8⁺ T cells and a decreased frequency of Treg cells.

We also demonstrated that PARP14 mediates IFNγ-driven progression in a model of spontaneous ICBT relapse. RNA-seq and GSEA indicated that the TME in tumours derived from YUMM2.1 that spontaneously progressed after α-PD-1 therapy was characterised by upregulated IFNγ-signalling and was T cell inflamed in contrast to the tumours derived from IFNγ pre-treated cells. This implies that chronic IFNγ-signalling may be responsible for both establishing T cell deserts, a feature of primary resistance, or conversely an immunosuppressive TME albeit T cell inflamed, but that in both instances this relies on the induction of PARP14. Our study focused on the ability of our therapeutics to influence cytotoxic T-cell responses; however, mounting evidence suggests that other immune cells are also important for α-PD-1 efficacy. For example, NK cells are essential for response to PD-1/PD-L1 blockade in some models[34]. Moreover, PARP14 facilitates the polarisation of macrophages into an M2-like state and of CD4⁺ T cells to the T$_{H}$2 lineage, both of which are cancer-promoting immune cell subtypes[14,15]. The effect of chronic IFNγ exposure on and possible contribution of PARP14 to tumour infiltration by myeloid and helper T cell populations also merits further investigation.

PARP14 appears to negatively regulate IFNγ signalling and responses, consistent with PARP14 being a negative feedback regulator of IFNγ signalling. Thus, we found that PARP14 pharmacological antagonism or silencing in tumour cells enhanced STAT1 phosphorylation and expression of STAT1 target gene products, including IRF1 responsible for the expression of many IFNγ signalling genes but also PARP14 itself[53,54]. We and others[15,55,56] show that PARP14 physically interacts with STAT1. In addition, PARP14 interacts with proteins that are commonly co-expressed following IFN treatment and which also interact with STAT1, thereby potentially regulating the IFN-induced interactome[55]. PARP9 is such a protein, which promotes STAT1 activation and pro-inflammatory gene expression in IFNγ-treated macrophages[15] as well as in pancreatic epithelial and cancer cells[57], although PARP9 has been shown to inhibit STAT1 induction of IRF1 in prostate cancer cells[58]. It will be interesting to determine whether PARP14 plays any role in converting PARP9 from a STAT1 co-activator into a STAT1 co-repressor. Significantly, RNA pol II ChIP-seq employed by Riley and colleagues identified 2744 genes down-regulated by PARP14, including STAT1 target genes encoding components of the antigen processing and presentation machinery[13]. As an alternative for

how PARP14 regulates STAT1, Iwata and colleagues suggested that PARP14 suppresses STAT1 by ADP-ribosylating STAT1 to prevent its phosphorylation in response to IFNγ[15]. Intriguingly, JAKi treatment was as effective as PARP14i treatment in restoring tumour immune suppression in tumours derived from YUMM2.1 that spontaneously progressed after α-PD-1 therapy. This implies that either decreasing STAT1 activity by JAKi to block expression of immunosuppressive molecules (including PARP14) or boosting STAT1 by PARP14i to enhance tumour immunogenicity are both viable approaches to increasing response to α-PD-1 therapy.

The combination of PARP inhibitors with ICBT has recently emerged as a promising therapeutic strategy for cancer, with PARP1/2/3 inhibitors, such as the FDA-approved drugs niraparib, olaparib, and rucaparib now at the forefront of clinical investigations[59,60]. It is becoming increasingly evident that the anti-cancer effects of PARP inhibitors go beyond their direct cytotoxic effects and that these drugs may also enhance α-PD-1 efficacy by activating the stimulator of interferon genes (STING) independently of BRCA status[61]. This is achieved through the generation of cytosolic double-stranded DNA fragments, which bind cyclic GMP-AMP synthase (cGAS) and activate STING, thereby inducing a type I IFNα/β response. Consequently, chemokine secretion and subsequent T cell infiltration are enhanced[61]. Notably, the PARP1 inhibitor talazoparib (BMN 673) elicited increased numbers of peritoneal CD8⁺ T cells and NK cells in an ovarian cancer mouse model and increased infiltration into ex vivo spheroids, in addition to increasing IFNγ and TNFα production levels[62,63]. The finding that PARP inhibitors upregulate PD-L1 in cancer cells further supports the rationale of combining PARP inhibition with α-PD-1 therapy for the treatment of ovarian, breast, and non-small cell lung cancer[64]. In addition to these established FDA-approved treatments, targeting other PARP family members can also enhance anti-cancer immune responses. The demonstration of both cancer cell-autonomous effects and anti-tumour immunity induced by enhanced IFN-signalling upon application of a highly selective PARP7 inhibitor (RBN-2397)[65], led to a phase 1 clinical study (NCT04053673) of this drug for patients with advanced solid tumours. Indeed, Falchook and colleagues showed upregulated *CXCL10* mRNA levels in tumours of patients treated with RBN-2397, accompanied by an increase in CD8⁺ GZMB⁺ T cells[66].

In conclusion, we propose antagonism of PARP14 with a catalytic inhibitor as a safe and efficacious approach to target high tumour cell intrinsic IFNγ signalling resulting from adaptation to α-PD-1 and thereby restore sensitivity to ICBT in progressing tumours.

## Methods

### Mouse tumour implant study design

Mice were housed in the Biological Services Facility of The University of Manchester on a 12/12 h light/dark cycle, and given unlimited access to food (Bekay, B&K Universal, Hull, UK) and water. All procedures were approved by the University of Manchester's animal welfare eithical review board and performed under relevant Home Office licences according to the UK Animals (Scientific Procedures) Act, 1986. Female, 8–12-week-old C57BL/6 or BALB/c mice were purchased from ENVIGO and allowed at least 1-week to acclimatise. YUMM2.1 cells ($7 \times 10^6$ cells), CT26 ($1 \times 10^6$ cells), and MC38 cells ($3 \times 10^5$ cells) in

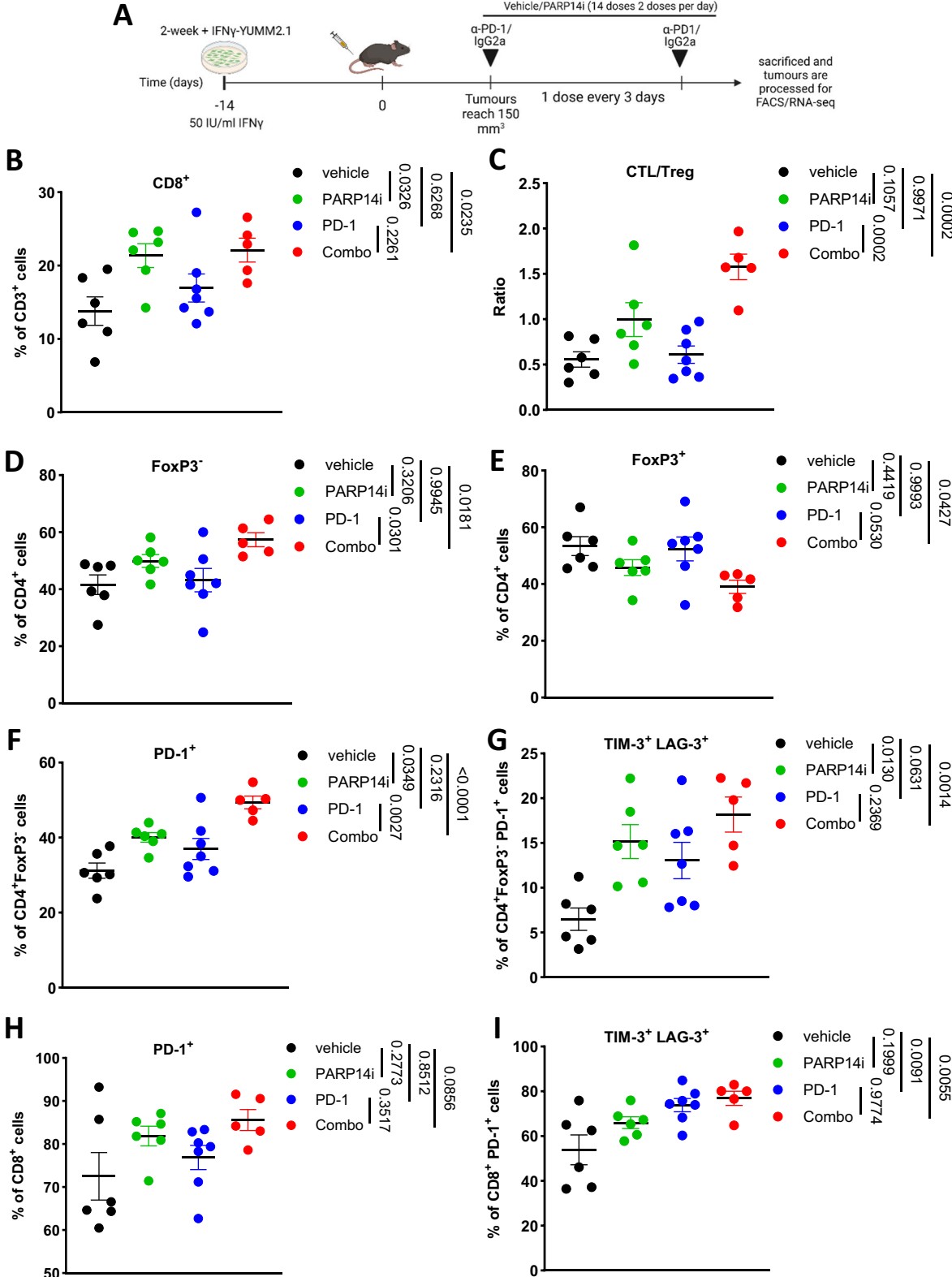

**Fig. 6 | Combination treatment with α-PD-1 and PARP14 inhibition remodels the tumour immune microenvironment. A** 8–12-week-old wild-type C57BL/6 mice were subcutaneously implanted with chronic IFNγ pre-treated YUMM2.1 cells. Treatment with either α-PD-1 or IgG2a antibody was initiated once tumour volume reached 150 mm³ (two doses, three days apart) and PARP14i or vehicle (two doses daily for a week). At the end of the treatment period, tumours were dissected and disaggregated for analysis by flow cytometry and RNA-seq. **B–I** Populations of

**B** CD8⁺ T cells, **C** the ratio of CD8⁺ GzmB+ cytotoxic lymphocytes (CTLs) to Tʳᵉᵍ cells, **D** CD4⁺ effector T cells (CD4⁺ FoxP3⁻), **E** Tʳᵉᵍ cells (CD4⁺ FoxP3⁺), PD-1⁺ of **F** CD4⁺ and **H** CD8⁺ T cells, and triple positive PD-1, TIM-3, LAG-3 of **G** CD4⁺ and **I** CD8⁺ T cells in the tumour infiltrate treated with different conditions (vehicle: *n* = 6; PARP14i: *n* = 6; α-PD-1: *n* = 7; Combo: *n* = 5). The data were presented as mean ± S.E.M. and the adjusted *p*-values were determined by one-way ANOVA Šídák's test. Source data are provided as a Source Data file.

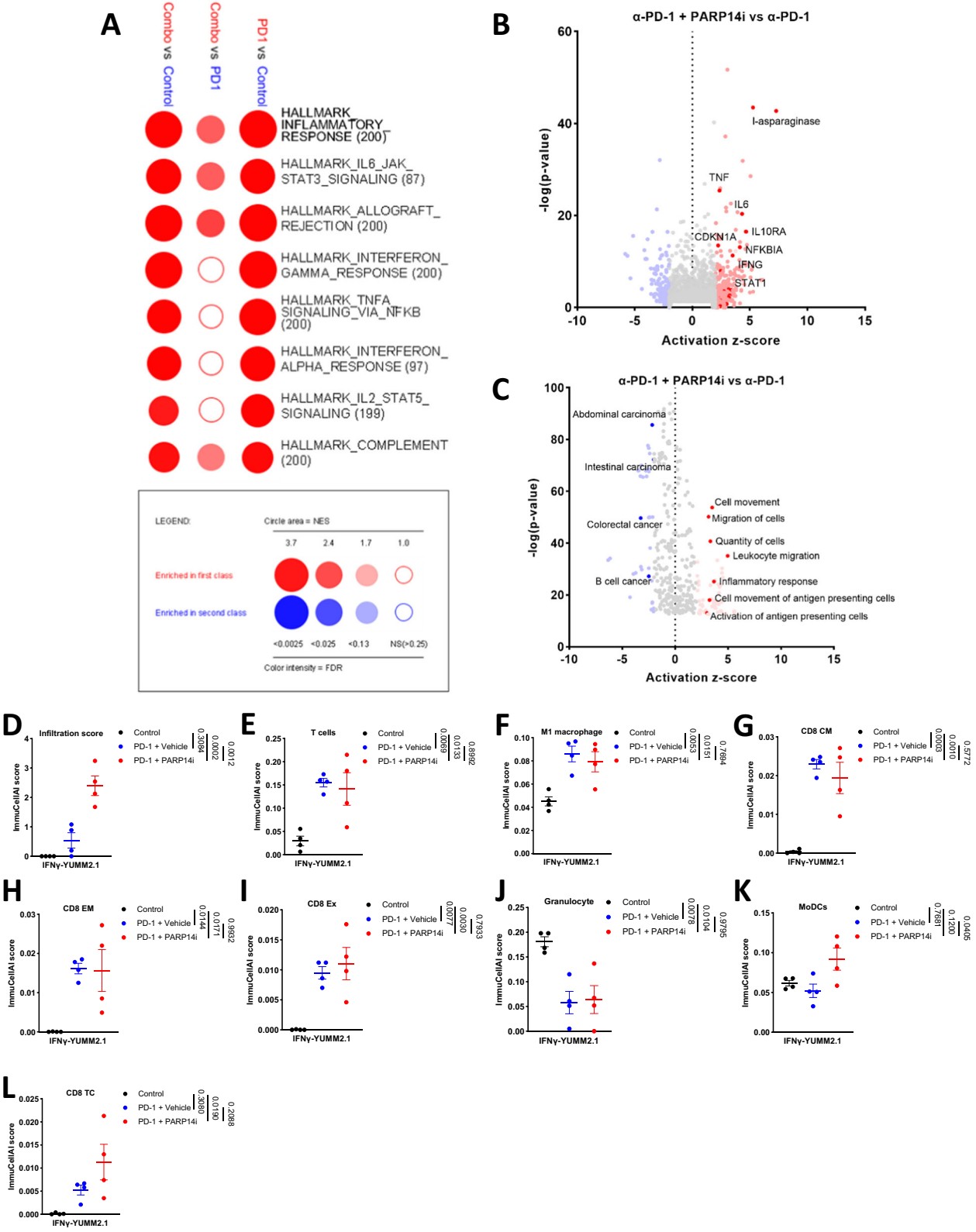

100 μL serum-free RPMI-1640 were subcutaneously injected into the left flank of mice under isoflurane anaesthesia. Tumour size (calculated by multiplication of height, width, and length calliper measurements) and mouse weight were monitored three times per week (every 2–3 days). When tumours reached an average volume of 80–100 mm³, mice were administered with up to four doses of 300 μg of α-PD-1 antibody (BioXCell) or rat isotype control antibody IgG2a (BioXCell) in

100 μL InVivoPure pH 7.0 Dilution Buffer (BioXCell) via intraperitoneal (i.p.) injection administered at 3–4-day intervals. Mice were also administered vehicle or 500 mg/Kg of RBN012759 by oral gavage twice a day (BID). RBN012759 was dissolved in 0.5 % w/v methylcellulose (Sigma–Aldrich) + 0.2 % v/v Tween 80 (Sigma–Aldrich). Each dose was delivered in a volume of 0.2 mL/20 g mouse (10 mL/kg) and adjusted for the last recorded weight of individual animals. Mice were

**Fig. 7 | Combination treatment with α-PD-1 and PARP14 inhibition induces an inflammatory response. A** GSEA based on RNA-seq data depicting hallmark processes enriched in chronic IFNγ pre-treated tumours treated with α-PD-1 alone ($n = 4$) versus Control ($n = 4$); combined αPD-1 and PARP14 inhibition (Combo) ($n = 4$) versus α-PD-1 alone ($n = 4$); Combo ($n = 4$) versus Control ($n = 4$). Circle area depicts the NES, and colour intensity depicts the FDR, with ≤0.25 classed as significant. **B–C** Ingenuity Pathway Analysis (IPA) was performed to identify up- or down-regulation of **B** upstream regulators and **C** disease-related or functional pathways in tumours receiving α-PD-1/PARP14i combination treatment ($n = 4$) versus α-PD-1 alone treatment ($n = 4$). Results were displayed with their $P$-value ($-\log(P\text{-value})$) and activation z-score. The $p$-values were assessed by two-tailed unpaired $t$-test. **D–L** Bulk-tumour RNA-seq results treated with Control ($n = 4$); PD-1 + Vehicle ($n = 4$); or PD-1 + PARP14i ($n = 4$) analysed by cell-type enrichment analysis (ImmuCellAI), with scores shown for **D** infiltration, **E** T cell, **F** M1 macrophage, **G** CD8 central memory (CM) T cells, **H** CD8 effector memory (EM) T cells, **I** exhausted (Ex) CD8 T cells, **J** Granulocytes, **K** Myeloid Dendritic cell (MoDC), **L** CD8 cytotoxic T cell (Tc). The data were presented as mean ± S.E.M. and the adjusted $p$-values were determined by one-way ANOVA Tukey's test. Source data are provided as a Source Data file.

monitored and body weight was measured daily. Mice were administered vehicle or 60 mg/Kg of Ruxolitinib (MedChemExpress) by i.p. injection once a day (QD). Ruxolitinib was dissolved in 0.5 %w/v methylcellulose (Sigma–Aldrich) + 0.1 %v/v Tween 80 (Sigma–Aldrich). Each dose was delivered in a volume of 0.1 mL/20 g mouse (10 mL/kg) and adjusted for the last recorded weight of individual animals. For CD8 T-cell depletion in vivo study, mice were administered with 5 doses of 100 μg of α-CD8α antibody (BioXCell) or rat isotype control antibody IgG2b (BioXCell) in 100 μL InVivoPure pH 7.0 Dilution Buffer (BioXCell) via intraperitoneal (i.p.) injection administered at 3–4-day intervals. Mice were culled once tumours reached 800 mm³, our predetermined experimental endpoint aligning with the principles of the 3Rs (Replacement, Reduction and Refinement) for improving animal welfare. In some cases, this limit has been exceeded the last day of measurement and the mice were immediately euthanised. A sample size of $n \geq 3$ per group was used throughout to achieve a statistical significance of $P < 0.05$. Mice were randomised into treatment groups. All tumours were included for analysis. Differences in survival were determined using the Kaplan–Meier method, and the $P$-value was calculated by the log-rank (Mantel-Cox) test using GraphPad Prism version 9.0c.

### Reagents
Recombinant human IFNγ (PHC4031) and recombinant mouse IFNγ (PMC4031) were purchased from Gibco and used at the indicated concentrations. The PARP14 inhibitors RBN012759[22] and RBN012811[31] were generated in-house and used at the indicated concentrations.

### Cell lines
The human melanoma cell lines A375, LOX-IMVI and 501-Mel (provided by Claudia Wellbrock, The University of Manchester), Lenti-X 293 T cells (provided by Angeliki Malliri, The University of Manchester), 5555, B16-F10, and MC38 cells (provided by Santiago Zelenay, The University of Manchester) and YUMM2.1 cells (provided by Richard Marais, The University of Manchester) were maintained in RPMI-1640 (Sigma–Aldrich) supplemented with 10 %v/v foetal bovine serum (FBS; Life Technologies) and 1 %w/v penicillin-streptomycin (P/ S; Sigma–Aldrich). All cells were maintained under standard conditions at 37 °C in a 5 %v/v CO₂ humidified incubator and passaged before reaching confluency. Cell lines were authenticated by STR profiling and cultures were routinely tested for mycoplasma contamination by PCR and deemed to be uninfected.

### Cell proliferation
Cells were fixed and stained with 0.5 %w/v crystal violet (Sigma) in 4 % w/v paraformaldehyde (PFA) in phosphate-buffered saline (PBS) for at least 30 min. Fixed cells were solubilised in 2 %w/v sodium dodecyl sulphate (SDS) in PBS and absorbance was measured at 595 nm using Biotek Synergy™ H1 Hybrid Multi-Mode Reader.

### Gene silencing
For siRNA-mediated silencing of STAT1, cells were seeded in 6-well plates (5 × 10⁵ cells/well) and incubated overnight. The next day, cells

were transfected with siRNAs using lipofectamine RNAiMAX (Life Technologies) transfection reagent according to the manufacturer's guidelines. After eight hours of incubation with the transfection mixture, the cell culture medium was replaced, and cells were incubated for 1–3 days at 37 °C. For shRNA-mediated silencing of PARP14, Lenti-X 293 T cells were seeded in T75 flasks (5 × 10⁶ cells/flask) and incubated overnight. The next day, cells were transfected with 4.5 μg of the respective shRNA/overexpressing vector, 6 μg of psPAX2 (12260; Addgene), and 3 μg of pVSVg (8454; Addgene); FuGENE HD transfection reagent (Promega) was used for the transfections. All shRNAs were cloned in pLV-EGFP lentiviral transfer vectors (VectorBuilder); the shRNA sequences are listed in Supplementary Table 1. The next day, the medium was replaced with fresh complete growth medium, and cells were incubated overnight. The following day, virus-containing supernatants were harvested, centrifuged at 1000 r.p.m. for 5 min, and filtered through 0.45 μm porous membranes (STARLAB). Lentiviral transductions were performed in a 6-well plate format (3 × 10⁵ cells/ well) using 10 μg/mL polybrene (Merck Millipore). Stably transduced cells were flow-sorted.

### Luciferase reporter assays
A375 melanoma cells were transfected with a pEZX-PG04 plasmid expressing Gaussia luciferase (GLuc) under the influence of the *PARP14* promoter (GeneCopoeia). After 48 h, 5000 cells were seeded in triplicate in 96-well plates and were treated with increasing concentrations of IFNγ for 24 h. Subsequently, 100 μL of supernatant was collected for further analysis, and the plate was stained with crystal violet for normalisation. The luminescence assay was carried out using the Genecopoeia Secrete Pair™ Dual Luminescence Assay kit according to the manufacturer's instructions. Luminescence was measured on a Biotek Synergy™ H1 Hybrid Multi-Mode Reader with normalisation to the crystal violet absorbance values.

### Quantitative real-time PCR (RT-PCR)
Total RNA was extracted using QIAzol Lysis Reagent and isolated using RNAeasy mini kit (both from QIAGEN). cDNA synthesis was performed using the ProtoScript II First Strand cDNA Synthesis Kit (NEB). Expression levels of target genes were determined by RT-PCR using SensiMix SYBR No-Rox (Bioline) with the primers shown in Supplementary Table 1. Reactions were run on an MX3000P real-time thermal cycler (Agilent Technologies) and Ct values determined using MxPro qPCR software. Relative expression levels were calculated using the $2^{-\Delta Ct}$ method after normalising to the expression levels of the housekeeping gene, *Gapdh*. Fold change levels were calculated using the $2^{-\Delta \Delta Ct}$ method after normalising to the untreated control.

### Western blot
Total proteins were extracted using SDS lysis buffer (4 %w/v SDS; 20 % v/v glycerol; 0.004 %w/v bromophenol blue; 0.125 M Tris-Cl, pH 6.8; 10 %v/v 2-mercaptoethanol) and sonication (50 kHz for 30 s; VibraCell X130PB, Sonics Materials) at 4 °C and subsequently denatured at 95 °C for 5 min. Proteins were separated on RunBlue 4-12 %w/v bis-tris polyacrylamide gels (Expedeon) and then transferred onto iBlot PVDF membranes (ThermoFisher) using the Wet/Tank Blotting Systems (Bio-

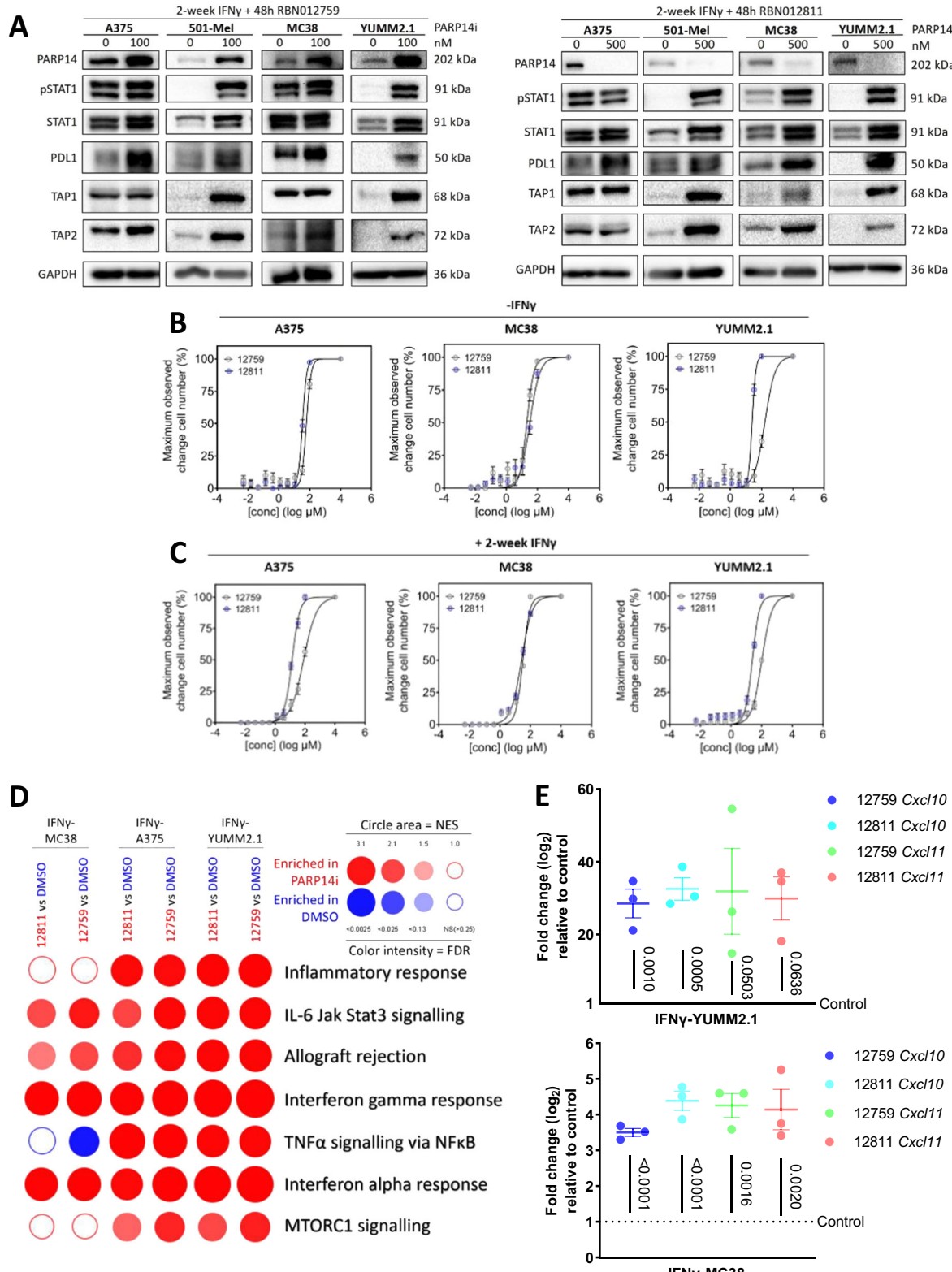

**Fig. 8 | PARP14 is a negative feedback regulator of IFNγ signalling. A** Chronic IFNγ pre-treated A375, 501-Mel, MC38, or YUMM2.1 cells were treated for 48 h with PARP14 pharmacological inhibitors RBN012759 (left) and RBN012811 (right). Expression of PARP14, pSTAT1, STAT1 and STAT1 target proteins is shown by western blot, with GAPDH visualised as a loading reference. **B–C** Graphs showing the relative maximum observed change in cell number as determined by crystal violet following 48 h treatment with varying concentrations of PARP14 inhibitor (12759: $n = 9$ per concentration; 12811: $n = 9$ per concentration) without IFNγ (**B**) or with 2-week IFNγ (**C**). The data were presented as mean ± S.E.M. **D** GSEA based on RNA-seq data depicting hallmark processes enriched in chronic IFNγ pre-treated tumours treated with RBN012759 or RBN012811 versus control (DMSO). The circle area depicts the NES, and colour intensity depicts the FDR, with ≤0.25 classed as significant. **E** RT-qPCR analysis of *Cxcl10* and *Cxcl11* mRNA expression levels in IFN-γ-YUMM2.1 and IFN-γ-MC38 cells treated with DMSO ($n = 3$); 12759 ($n = 3$); 12811 ($n = 3$). The data were presented as mean ± S.E.M. and the adjusted *p*-values were assessed by one-way ANOVA Dunnett's test. Source data are provided as a Source Data file.

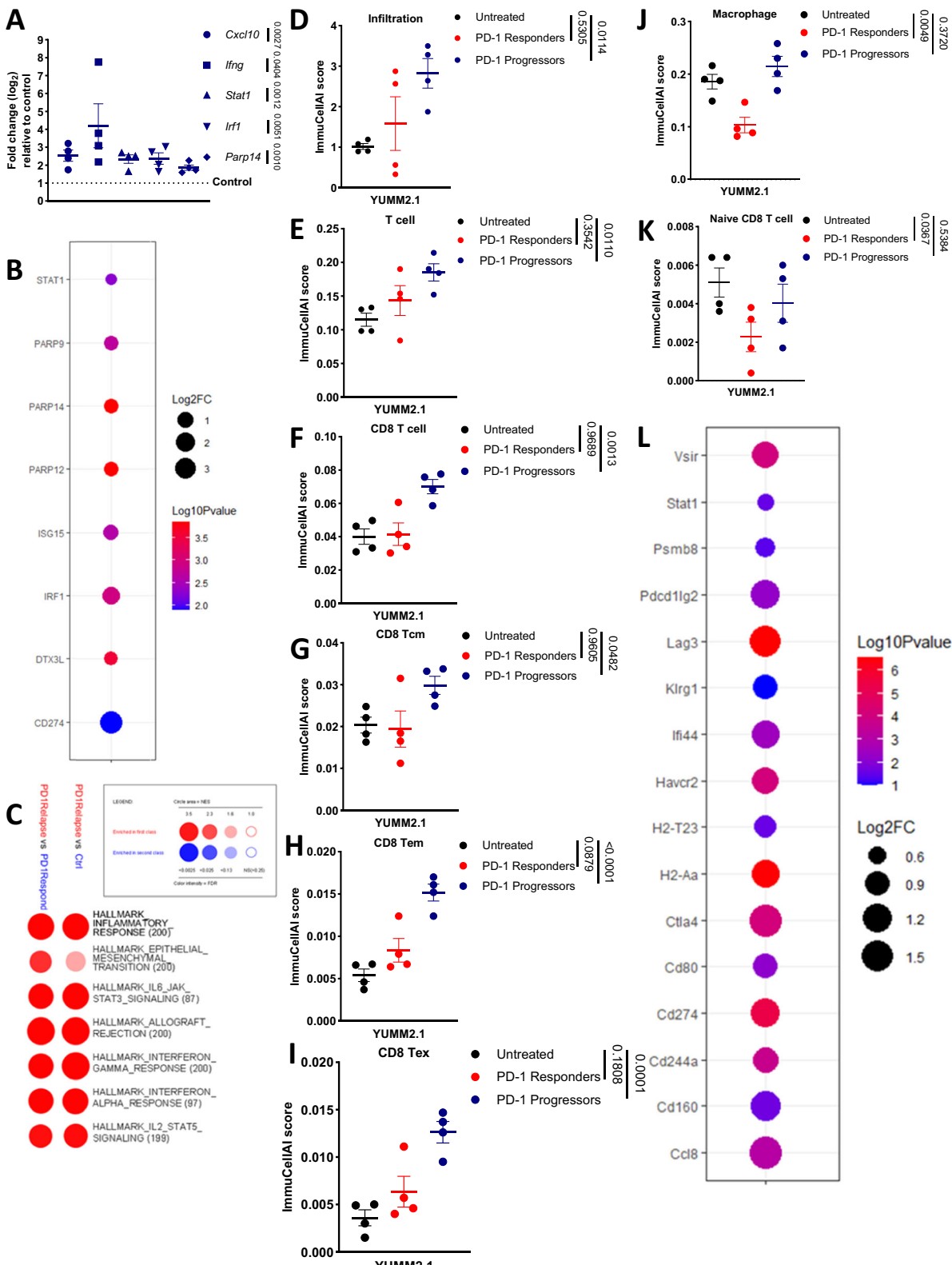

Rad). Membranes were probed overnight at 4 °C in blocking solution containing the primary antibody. Primary antibodies used in this work were PARP14 (C-1) (Santa Cruz Biotechnology, sc-377150), Phospho-Stat1 (pSTAT1; Tyr701; 58D6) (Cell Signalling Technology, 9167 L), STAT1 (Cell Signalling Technology, 9172), MHC Class I H2 Kb (Abcam, ab93364), GAPDH (Proteintech, 60004-1-Ig), TAP1 (Cell Signalling Technology, 12341), TAP2 (Cell Signalling Technology, 12259; Santa

Cruz Biotechnology, sc-515576), and PD-L1/B7-H1 (R&D Systems, AF1019-SP; Cell Signalling Technology, 13684). This was followed by incubation with the appropriate secondary antibody for 1.5 h at room temperature. Signals were developed using the Clarity Max Western ECL blotting substrate (Bio-Rad) and acquired on a Gel Doc XR+ Gel Documentation System (Bio-Rad). Images were analysed using Image Lab™ Software (version 3.0.1.).

**Fig. 9 | YUMM2.1 tumours spontaneously relapsing after α-PD-1 treatment are highly inflamed with T cells. A** RT-qPCR analysis of *Cxcl10*, *Ifng*, *Stat1*, *Irf1*, and *Parp14* mRNA expression levels in YUMM2.1 tumours which (relapsed after α-PD-1 antibody treatment: *n* = 4) compared to IgG2a-treated tumours (control: *n* = 4). The data were presented as mean ± S.E.M. and the *p*-values were assessed by two-tailed unpaired *t*-test. **B** Differential gene expression of IFNγ targent genes in short-term melanoma cell cultures derived from ICBT-progressing lesions with elevated IFNγ-signalling compared to cultures where intrinsic IFN-signalling was minimal. Circle area depicts-log$_2$ fold change and colour intensity depicts-log$_{10}$(*p*-value), with ≥1.30 classed as significant. The *p*-values were assessed by two-tailed unpaired *t*-test. **C** GSEA based on RNA-seq data depicting hallmark processes enriched when comparing YUMM2.1 tumours that relapsed after α-PD-1 treatment versus control; and relapsing after α-PD-1 treatment versus responding during α-PD-1 treatment.

Circle area depicts the NES, and colour intensity depicts the FDR, with ≤0.25 classed as significant. **D–K** Bulk-tumour RNA-seq results derived from Untreated (*n* = 4), PD-1 Responders (*n* = 4), and PD-1 Progressors (*n* = 4) analysed by cell-type enrichment analysis (ImmuCellAI), with scores shown for **D** infiltration, **E** T cell, **F** CD8 T cell, **G** CD8 Tcm, **H** CD8 Tem, **I** CD8 Tex, **J** Macrophage, and **K** Naive CD8 T cell. The data were presented as mean ± S.E.M. and the adjusted *p*-values were assessed by one-way ANOVA Šídák's test. **L** Expression differences of immuno-suppressive gene in tumours derived from α-PD-1-relapsing condition compared with control condition by sequencing bulk mRNA. Circle area depicts-log$_2$ fold change and colour intensity depicts -log$_{10}$(*p*-value), with ≥1.30 classed as significant. The *p*-values were assessed by two-tailed unpaired *t*-test. Source data are provided as a Source Data file.

## RNA sequencing

For cell models treated continuously for 2 weeks with BSA or IFNy, total RNA was isolated from triplicate samples using the RNAeasy mini kit. RNA integrity was assessed on an Agilent 2200 TapeStation (Agilent Technologies). RNA samples (~1 µg) were submitted for RNA sequencing (100 nt paired-end reads, <30 million reads per sample) using an Illumina HiSeq4000. Three samples per condition were sequenced. Sequence data was collected using Hiseq Software Suite (version 3.4.0). Read quality was assessed using FastQC. Raw reads were trimmed using trimmomatic (version 0.36.6; sliding window trimming with 4 bases averaging and average quality minimum set to 20). Trimmed reads were aligned to the reference genomes hg38_analysisSet (human) or mm10 (mouse) using HISAT2 (version 2.1.0; default parameters). Aligned reads were counted against GENCODE release 25 (human) or GENCODE release M14 (mouse) using htseq-count (version 0.9.1). Differential gene expression analysis was performed using edgeR (version 3.24.1). Heatmap generation and clustering were performed using Multiple Experiment Viewer (version 10.2).

For the in-depth time-course analysis, RNA was isolated from triplicate YUMM2.1 and CT26 cell samples stimulated chronically with IFNg and rechallenged weekly with fresh IFNg for 2 and 24 h using the Qiagen RNeasy mini kit (catalogue #74106) per manufacturer's instructions. RNA concentration was determined using a NanoDrop 8000 (Thermo-Fisher Scientific, Waltham, MA). Paired-end sample libraries consisting of 60 bp with 6 nucleotide indices were prepared for measuring high-throughput 3′ digital gene expression based on a previously published protocol (Massachusetts Institute of Technology, Cambridge, MA). RNA-seq read counts were imported into R using tximport (v1.28.0). The vst method in DEseq2 was used to variance-stabilise read counts, select the 5000 most variable genes and, together with PCA, identify outlier samples, resulting in one YUMM2.1 and four CT26 samples being discarded prior to downstream analyses. To derive a "chronic IFNγ signature", we first derived a separate signature for YUMM2.1 or CT26 cell lines. Our approach was to identify genes upregulated following 3-weeks of chronic IFNγ stimulation compared to baseline (Log2-FC > 0 and p-adjusted <0.1) and to eliminate genes that responded either to IFNγ rechallenge or that were upregulated in parallel cultures not treated with IFNγ. The final sixteen-gene consensus chronic IFNγ signature was obtained by retaining the overlapping genes of chronic signatures for both YUMM2.1 and CT26 cell lines. The association of our chronic IFNγ signature with survival in ICBT cohorts was assessed using Tumour Immune Dysfunction and Exclusion (TIDE) website algorithms (http://tide.dfci.harvard.edu/login/)[67]. To test for association with patient survival of historic cohorts, Kaplan–Meier survival analysis of TCGA data was performed using Gene Expression Profiling Interactive Analysis (GEPIA2) (http://gepia2.cancer-pku.cn/#index)[68].

## Gene set enrichment analysis (GSEA) and Ingenuity pathway analysis

GSEA using BubbleGUM software (version 1.3.19) was performed using "BubbleMap" settings[24], selecting the Broad Institute Molecular

Signatures Database (MSigDB) version 7.0. The "MaxMean" test statistic was used to test enrichment using a two-class comparison. Genes were ranked based on the signal-to-noise ratio. All *P*-values and false discovery rates (FDR) were based on 500-1,000 permutations. Pathway overrepresentation analysis was conducted using Ingenuity Pathway Analysis[69] (QIAGEN; version 01-12). Genes used for pathways were pre-filtered to remove lowly expressed genes.

## Tumour immune infiltrate analysis by flow cytometry

When tumours reached the required endpoint volume, mice were sacrificed by cervical dislocation, and tumours were dissected. Tumours were incubated for 45 min with 100 µg/mL of Liberase (Sigma–Aldrich) in serum-free media at 37 °C and then pushed through a BD Falcon 100 µM nylon cell strainer using a syringe plunge. The cell suspension was centrifuged at 1300 r.p.m. and 4 °C for 7 min, and cells were stained for 20 min protected from light with LIVE/DEAD™ Fixable Blue Dead Cell Stain Kit (ThermoFisher) diluted at 1:1000 in PBS. Subsequently, Fc receptors were blocked, and cells were stained with surface stain antibody mix for 45 min at 4 °C in the dark. Cells were fixed and permeabilised using the Foxp3/Transcription Factor Staining Buffer Set (eBioscience) following the manufacturer's instructions. Intracellular staining was performed for 45 min at 4 °C in the dark, after which samples were measured on a BD FACSymphony flow cytometer (BD Biosciences) and data collected using BD FACSDiva™ software. For all antibodies, a non-stained cell sample and appropriate fluorescence minus one control were analysed as well. Data were analysed using FlowJo version 8.7.

## Antibodies and reagents for flow cytometry

The following antibodies were purchased from BioLegend: CD16/32 (clone 93), CD45 (clone 30-F11), CD3 (clone 145-2C11), CD4 (clone GK1.5), CD8α (clone 53-6.7), CD25 (clone PC61), CD62L (clone MEL-14), CD69 (clone H1.2F3), PD-1 (clone 29 F.1A12), CD44 (clone IM7), Granzyme B (clone QA16A02), LAG-3 (clone C9B7W), TIM-3 (clone RMT3-23), TCR γ/δ (clone GL3), NK1.1 (clone PK136), and TNF-α (clone MP6-XT22), LAP (clone TW7-20B9), IL-10 (clone JES5-16E3), and Ki-67 (clone 11F6). FOXP3 (clone FJK-16s) and TCR beta (clone H57-597) were purchased from eBioscience.

## Inhibiting PARP14 in T cells in vitro

T cells were isolated from BALB/C mice between 8–16 weeks of age. Briefly, spleens were collected and a single-cell suspension was prepared by mechanical disruption, by pushing the organs through a 70 µm cell strainer (Falcon) using a plunger to push tissue through. Cells were pelleted at 500 × g for 5 min, followed by red blood lysis by incubating cells in RBC lysis buffer 1X (Sigma–Aldrich, R7757-100ML) for 10 min at room temperature. MojoSort™ Buffer (BioLegend; 480017) was added and the cell suspension was pelleted at 500 × g, room temperature for 5 min. After cells were counted, they were resuspended to in 1 mL of MojoSort™ Buffer per 1 × 10$^8$ total cells. CD3 + T cells were isolated by negative selection using the MojoSort™

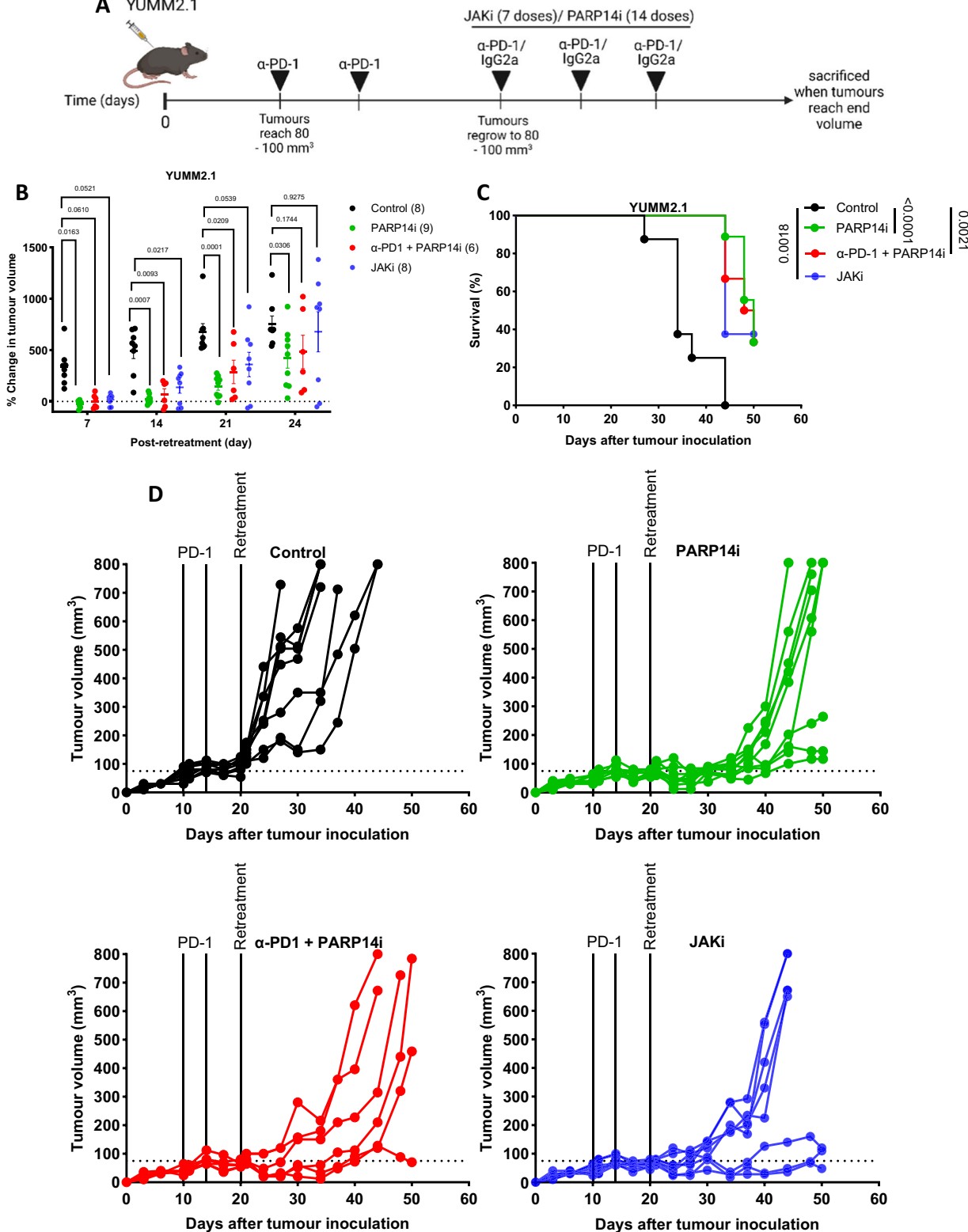

**Fig. 10 | Both PARP14 inhibition and JAK inhibition can improve the survival of mice bearing relapsing YUMM2.1 tumours. A** 8–12-week-old wild-type C57BL/6 J mice were subcutaneously implanted with YUMM2.1 cells. Treatment with α-PD-1 antibodies (two doses, three days apart) was initiated once tumour volume reached ~80 mm³. Once the tumours had regrown to roughly the size at which α-PD-1 treatment was initially commenced, they were randomised into four treatment groups: control (n = 8); PARP14i (14 doses; 2 doses per day) (n = 9); α-PD-1 antibodies (3 doses of α-PD-1 in every 3 days) and PARP14i (14 doses; 2 doses per day)

(n = 6); Ruxolitinib (7 doses; 1 dose per day) (n = 8). **B** The percentage change in tumour volume between the start of retreatment and day 7, 14, 21, and 24 post-retreatment. The data were presented as mean ± S.E.M. and the adjusted p-values were assessed by two-way ANOVA Tukey's test. **C** Kaplan–Meier survival plots for each treatment arm. The p-values were assessed by Log-rank (Mantel-Cox) test. **D** Growth curves for each tumour per condition. Source data are provided as a Source Data file.

Mouse CD3 T Cell Isolation Kit according to the instructions of the manufacturer (BioLegend; 480024). Isolated T cells were counted and resuspended in complete primary T cell growth medium. Cells were seeded at a density of $1 \times 10^6$ cells per mL and grown in primary T cell media supplemented with 3 μg/mL anti-CD3 (BioLegend; 100238) and anti-CD28 antibodies (BioLegend; 102116) and 1 μM DMSO (Sigma–Aldrich; 67-68-5) or PARP14i (Ribon Therapeutics; RBN012759-004). Cells were passaged and split with the refreshment of T-cell growth medium with 1 μM DMSO (Sigma–Aldrich; 67-68-5) or PARP14i (Ribon Therapeutics; RBN012759-004) until they became rested. Upon cells becoming rested, they were re-activated by seeding at a density of $1 \times 10^6$ cells per mL and grown in primary T cell media supplemented with 3 μg/mL anti-CD3 (BioLegend; 100238) and anti-CD28 antibodies (BioLegend; 102116) for either 14 h for cytokine staining or 96 h for transcription factor and differentiation factors staining by flow cytometry. Cells were incubated with Brefeldin A Solution (BioLegend; 420601) for 4 h before harvesting for cytokine staining.

## Co-IP
Cells were lysed in IP lysis buffer (Thermo Scientific; 87787) when their confluency reached 80%. Cell lysate was incubated with either STAT1 antibody (4 μg, Proteintech; 10144-2-AP) or isotype control IgG and Dynabeads™ Protein A for Immunoprecipitation (Thermofisher; 10001D) with rotation overnight at 4 °C, followed by washing three times with PBS/Tween 20 (0.02%), using a magnet to collect the beads after each wash. Fifteen per cent of the precipitated protein sample was subjected to SDS–PAGE. Visualisation was carried out using the Gel Doc™ XR+ Gel Documentation System (Bio-Rad).

## In silico immunophenotyping
Computational immunophenotyping was performed using Immune-CellAI analysis[30]. Statistical significance was determined by unpaired two-tailed Mann-Whitney test using GraphPad Prism (version 9.0).

## Statistical analysis
Statistical analysis calculations were carried out using Microsoft Excel and GraphPad Prism software 9.0. For each of the experiments, the statistical experiment was performed separately. $p$-values $< 0.05$ was considered statistically significant. All data was expressed in the form of mean ± std error unless otherwise specified.

## Reporting summary
Further information on research design is available in the Nature Portfolio Reporting Summary linked to this article.

## Data availability
Source data are provided with this paper. The RNA-seq data generated in this study have been deposited in the ArrayExpress and Gene Expression Omnibus (GEO) database under accession codes E-MTAB-12194, E-MTAB-12195, E-MTAB-12196, E-MTAB-12872, and GSE237098.

The TCGA data used are publicly available at the Genomic Data Commons portal (https://portal.gdc.cancer.gov/). Transcriptomic data —fragments per kilobase of transcript per million mapped fragments (FPKM) and transcripts per million (TPM) data— from pre-treatment and on-treatment biopsies of melanoma patients undergoing ICBT[32] were obtained from the GEO database with accession number GSE91061; also from melanoma cell cultures from ICBT-progressing lesions[7] in the Sequence Read Archive under accession code PRJNA818797. TPM values were converted to $\log_2(TPM + 1)$. ChIP-seq data for STAT1 were retrieved from the Encyclopaedia of DNA Elements (ENCODE) project database [https://www.encodeproject.org]. The remaining data are available within the Article, Supplementary Information or Source Data file. Source data are provided with this paper.

## Code availability
The workflow to build the chronic IFNγ gene signature is available from Zenodo with the identifier [https://doi.org/10.5281/zenodo.8322092].

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

## Acknowledgements

This study was funded by a research sponsorship agreement between Ribon Therapeutics and The University of Manchester, grant 821034 from the Melanoma Research Alliance (MRA), grant 22-0091 from Worldwide Cancer Research (WCR), and awards from the Skin Cancer Research Fund (SCaRF) and Melanoma UK, all to A.H, and from the Christie Charity Fund to P.L. C.W.W. was funded by a scholarship from the Hong Kong Scholarship for Excellence Scheme (HKSES). C.E. was funded by a studentship from the Medical Research Council (MRC). We thank Dr Bin Gui, Dr Roma Kaul, Dr Kaiko Kunii, and other employees of Ribon Therapeutics for helpful discussions and advice. We thank Dr Gareth Howell and The University of Manchester Flow Cytometry Core Facility for facilitating flow cytometry analysis, Dr Leo Zeef and the Genomics Technologies and Bioinformatics Core Facilities for assistance with RNA-seq, and members of the Biological Services Facility at The University of Manchester for help with animal work. We thank Michael Eisenstein for editing support. Figs. 1A, 3A, E, 4A, 5A, F 6A, 10A, and Supplementary Figs. 14A, B, F were created by BioRender.

## Author contributions

C.E., R.T.N and A.H. conceived the study. C.W.W., C.E., K.N.S., R.L., W.Z., V.G., S.C., K.S., M.L.F.C., E.U., H.M., B.A.T., C.L., D.T.I., N.R.P. and R.T.N. performed experiments. C.W.W., C.E., K.J.W., P.E.R., M.N., R.T.N. and A.H. contributed to experimental design and data analysis. A.H., D.J.W., M.P.S., K.M., K.J.W., P.E.R., R.T.N. and M.N provided supervision. C.W.W., C.E., K.N.S. and A.H. drafted the original version of this manuscript, with all authors reviewing subsequent drafts. P.L., M.N. and A.H. obtained funding for the study.

## Competing interests

The authors declare the following competing interests: D.T.I., C.L., N.R.P. and M.N are all employees and shareholders of Ribon Therapeutics at the time of data collection. P.E.R. served as a consultant to Ribon Therapeutics. A.H. received research sponsorship from Ribon Therapeutics. All other authors declare no competing interests.

## Additional information

¹Faculty of Biology, Medicine and Health, The University of Manchester, Manchester M13 9PT, UK. ²Lydia Becker Institute of Immunology, The University of Manchester, Manchester M13 9PT, UK. ³Cancer Data Science Laboratory, National Cancer Institute, Bethesda, MD 20814, USA. ⁴Ribon Therapeutics Inc., 35 Cambridge Park Drive, Suite 300, Cambridge, MA 02140, USA. ⁵Department of Medical Oncology, The Christie NHS Foundation Trust, Wilmslow Road, Withington, Manchester M20 4BX, UK. ⁶Patricia E. Rao Consulting, Acton, MA 01720, USA. ⁷Colorectal and Peritoneal Oncology Centre, The Christie NHS Foundation Trust, Wilmslow Road, Withington, Manchester, UK. ⁸These authors contributed equally: Kieran N. Sefton, Rotem Leshem. ✉e-mail: adam.hurlstone@manchester.ac.uk

