## [Peer Review File · Nature Communications]

PARP14 inhibition restores PD-1 immune checkpoint inhibitor response following IFN γ -driven acquired resistance in preclinical cancer modelsReviewers' comments:

Reviewer #1 (Remarks to the Author): with expertise in IFN signaling in cancer, cancer immunology/therapy

Both positive and negative roles of the IFN-g signaling in immunotherapy responses have been reported, which is likely determined by a multitude of factors, including the duration of IFN-g stimulation. To this end, the Minn group from UPenn reported that chronic stimulation with IFN-g induces immunotherapy resistance, via both PD-L1-dependent and -independent mechanisms (i.e, STAT1-mediated epigenetic changes). Given the pleiotropic functions of the IFN-g signaling, other underlying mechanisms likely exist. Here, Wong, et al. explored this further in an attempt to identify actionable targets for immunotherapy resistance resulting from chronic IFN-g exposure. Using various tumor models for melanoma and colorectal cancer, they found that prolonged IFN-g pretreatment upregulated poly-ADP ribosyl polymerase 14 (PARP14) in tumor cells, coupled with anti-PD-1 resistance. Clinically, they found a positive correlation of PARP14 with IFNG in human skin cutaneous melanomas (SKCMs) from the TCGA database and that PARP14 expression was increased in patient samples treated with anti-PD-1. To test how PARP14 upregulation in tumor cells modulates anti-PD-1 resistance in tumors after chronic exposure to IFN-g, the authors knocked down PARP14 in tumor cells, which significantly improved mouse survival upon anti-PD-1. As a pharmacological approach, they treated mice bearing established IFN-g-pretreated tumors with RBN012759, an inhibitor for PARP14 (PARP14i). While PARP14i alone only suppressed YUMM2.1 melanoma but not CT26 or MC38 colorectal tumors, when combined with anti-PD-1, it resulted in potent suppression of all these tumors, with 100% of YUMM2.1-bearing mice as well as ~60% of mice bearing either CT26 or MC38 surviving to the end of experiments (60 days for melanoma, 55 days for CT26, and 35 days for MC38). Clearly, these results indicate that PARP14i resensitized these anti-PD-1-resistant tumors to anti-PD-1, leading to strong synergistic effects. To determine how PARP14 in host cells contributes to this resensitizing effect, the authors observed that PARP14 KO mice bearing YUMM2.1 melanoma responded better to anti-PD-1, suggesting host cell PARP14 is also important in the resensitization to anti-PD-1. They further demonstrated that CD8+ T cells were essential for therapeutic efficacy of PARP14i+anti-PD-1. While these data imply that PARP14 in both tumor and host cells regulates anti-tumor immunity and PARP14i boosted anti-PD-1 efficacy, overall data quality and significance need to be substantially improved.

Major Concerns

1. As aforementioned, immunotherapy resistance associated with chronic IFN-g treatment has been reported by the Minn group, who also showed that direct inhibition of JAK1/2 with Ruxolitinib can overcome the immunotherapy resistance. This greatly dampens my enthusiasm and overall significance of this study.
2. As demonstrated here and by the Minn group, chronic activation of the IFN-g signaling in tumor cells is detrimental to anti-PD-1 therapy. Why would further activation of IFN-g signaling with PARP14i be able to resensitize IFN-g-pretreated tumors to anti-PD-1? Also, this is opposite to previous reports showing JAK1/2i is effective at overcoming immunotherapy resistance in IFN-g chronically treated melanoma.
3. The inconsistency of the models used, data reproducibility, and scientific rigor. For instances, naïve tumor cells without prolonged IFN-g pretreatment (Fig. 2A-C) were used in the PARP14KD and PAPP14 KO mouse experiments (Fig. 2G-I), but IFN-g pretreated tumor cells were used in the experiments with PAPP14i (Fig. 2D-E and supplemental Fig. 4); also, different treatment regimens were used for the immunoprofiling studies in Fig. 3C and therapeutic studies in Fig. 2D; in addition, different ending times were chosen for YUMM2.1 (60 days), CT26 (55 days), and MC38 (35 days). All the experiments were done just once and not both sexes of mice were used, which is not compliant with the requirements of Nat. Comms.
4. The causal relationship of PARP14 upregulation in tumor cells and host cells to anti-PD-1 resistance (or IFN-g resistance) remains to be established. As discussed above, IFN-g pretreated tumor cells (anti-PD-1-resistant) should be used in PARP14KD and PAPP14 KO mouse experiments, like what they did with the PARP14i. PARP14 should be re-expressed in IFN-g-pretreated PARP14KD cells to establish a direct link of tumor PARP14 expression to anti-PD-1 resistance. While their human data of a correlation between PARP14 and IFNG indicated that IFN-g (Fig. 1I) and anti-PD-1 (supplemental Fig. 7E) upregulated PAPP14, they nevertheless didn't establish that PARP14 expression in human tumor cells drives resistance to either anti-PD-1 or IFN-g. On an extended note, PARP14 should be knocked out with CRISPR-Cas9 to explicitly assess tumor PARP14 in anti-PD-1 resistance, unless this would be lethal to tumor cells.
5. The relative importance of tumor vs host expression of PARP14 awaits further investigations. Their data support that PARP14KD in naïve (not IFN-g pretreated) tumor cells can prevent chronic IFN-g treatment-induced anti-PD-1 resistance, but they don't show that PARP14KD can remedy anti-PD-1 resistance after chronic IFN-g treatment. On the other hand, deletion of CD8+ T cells completely abolished the efficacy of combined PARP14i+anti-PD-1 therapy in overcoming

anti-PD-1 resistance in IFN-g-pretreated YUMM2.1 melanoma. This, together with the results from PAPR14 KO mice, establishes an important role of PARP14 in CD8 T cells in PARP14i+anti-PD-1. Similar studies should be done with in vivo neutralizing antibodies against CD4+ T cells. Additionally, IFN-g-pretreated PAPR14KO tumor cells should be used in Fig. 2A-C to explicitly pinpoint the role of tumor expression of PAPR14 in this process.

Minor Concerns

1. What does “pretreatment” label on the X axis of Fig. 1B mean? Why is the label of the X axis in Fig. 1G different from that of Fig. 1F and 1H? Also, Fig. 2I was partially cut off.
2. Please provide the rationale to study PARP14 but not PARP9 or PARP12, given they were all upregulated by IFN-g. Further, PARP9 should be examined, given the functional antagonism between PARP14 and PARP9.
3. It is surprising that no long-term protection was observed in tumor-bearing mice with their rigorous regimen of anti-PD-1 (300 ug/mouse for 4 doses), given that these tumors are regarded as responsive tumor types to immunotherapy.
4. Knock-down of PARP14 did not seem to work well (Supplemental Fig. 3A).
5. Increased expression of PARP14 after RBN012759 may obliterate the therapeutic effects of PARP14i. RBN012811 may be a better option.
6. Fig. 5A showed upregulated PARP14 in anti-PD-1-resistant bulk tumor as compared to untreated bulk tumor. This should be compared to anti-PD-1-responsive bulk tumors. Also, analysis of PARP14 expression in tumor cells should be done.
7. Supplemental Fig. 7G was not mentioned in the text.
8. The gating strategies are not accurate.
9. Both forward and reverse primer sequences should be provided in supplemental table 1.
10. “Depleting PARP14” and “depletion of PARP14” were often used, but only knock-down of PARP14 was performed. In addition, the following sentences are confusing; please re-write.

“Contrasting IFNG and STAT1 mRNA abundance in pre- versus on-treatment melanoma biopsies also supported that IFN γ -signalling is induced by ICBT.”

“However, PARP14 mRNA levels could not predict the depth or duration of clinical response among responders to ICBT in the Riaz et al. cohort²⁸ (data not shown). This may reflect several confounding factors, including a limited number of samples available for analysis; highly

heterogeneous tumours in which only small regions were sampled; sampling of different lesions pre- and on-treatment; and finally, biopsies that were not necessarily sampled at the point of relapse, which is when our in vivo data indicate that PARP14 induction may be at its highest and exerting its greatest influence.”

Considering all these different factors, even a more robust longitudinal examination of PARP14 induction in patient biopsies undergoing α -PD-1 therapy may not provide a solution to the discrepancy between this study and the study by Riaz, et al., as it is not feasible to biopsy the tumor tissues from the same loci at different times.

Reviewer #2 (Remarks to the Author): with expertise in IFN signaling in cancer, cancer immunology/therapy

Wong et al. demonstrated that in vitro IFN γ -pretreated cancer cells transplanted into syngenic mice form tumors which are no longer responsive to anti-PD1 ICB, in contrast to tumors established from IFN γ -naive control cells. This was shown in YUMM2.1 melanoma and MC38 / CT26 colon carcinoma models. The authors recognized PARP14 among the upregulated genes in several IFN γ -pretreated tumor cell lines. Downregulation of PARP14 by shRNA (shPARP14) in IFN γ -pretreated YUMM1.2 cells sensitized corresponding tumors to anti-PD1 ICB. In vitro IFN γ -treated shPARP14-YUMM2.1 cells showed enhanced expression of STAT1 target genes and corresponding tumors displayed altered T cell infiltrates with higher CD8/Treg ratios. Moreover, the authors demonstrated that PARP14 inhibitors (PARP14i) sensitized IFN γ -pretreated YUMM2.1 tumors to anti-PD1 treatment (with similar observation in IFN γ -pretreated CT26 and MC38 tumor models). Important to note, experiments with PARP14 $^{-/-}$ knockout mice demonstrated that PARP14 deficiency in host cells also significantly improved the outcome of anti-PD1 ICB.

Comments of the reviewer:

Chronic IFN γ signaling in tumor cells has recently been linked to immunotherapy resistance. The underlying non-genetic mechanisms are still poorly understood but seem to be multifaceted. Here, Wong et al. propose involvement of PARP14 in this process. PARP14 (ARTD8) is known as negative regulator of STAT1 and inflammatory genes, but findings of the authors in the context of chronic IFN γ signaling in tumor cells are novel and of potential clinical interest. However, important data supporting their conclusions are missing.

Major concerns:

The authors summarize their findings in the discussion as follows: „... For our in vivo experiments, we employed YUMM2.1, CT26 and MC38 syngeneic mouse tumour models, which initially respond to α -PD-1 therapy but subsequently relapse, MC38 more quickly than YUMM2.1 or CT26. Chronic IFN γ treatment before implantation rendered tumours resistant from the outset to α -PD-1 therapy, but response to α -PD-1 could be restored by depleting PARP14 in tumour cells ...“

Comment: This is a general statement over all models on the role of tumor cell-intrinsic PARP14 in resistance to anti-PD1 therapy under chronic IFN γ exposure. However, in case of CT26, the role of tumor-intrinsic PARP14 has not at all been studied. According to the Western blot (suppl Fig. 1E/D, [labelling in the figure is not correct]), IFN γ -treated CT26 cells only barely upregulate PARP14, if at all. This raises concerns about the role of tumor cell-intrinsic PARP14 in anti-PD1 resistance of IFN γ -pretreated CT26 tumors. Does PARP14 CRISPR/Cas9 knock-out restore the anti-PD1 sensitivity of IFN γ -pretreated CT26 tumors? Can effects of PARP14i on IFN γ -pretreated CT26 tumors just be explained by its effects on host cells?

shPARP14 YUMM2.1 and shPARP14 MC38 cells show only modest decrease in PARP14 protein expression (suppl. Fig. 3a). The authors themselves define the effects of shPARP14 downregulation on anti-PD1 sensitivity of IFN γ -pretreated MC38 tumors as “less robust” (suppl Fig. 3D). In order to obtain clear results PARP14 should be knocked out (CRISPR/Cas9) in all models.

In Figure 5B/C the authors show that tumours which spontaneously relapsed following α -PD-1 therapy could be re-sensitised by PARP14i treatment. Does it have a clear impact also on survival?

Does combined anti-PD1 plus PARP14i treatment have an effect on tumor growth in IFN γ -naive models? It is expected that in anti-PD1 sensitive tumor models (YUMM2.1, CT26, MC38), IFN γ is released early after therapy onset, suggesting that early blockade of PARP14 activity by selective inhibitors should prevent or delay development of adaptive resistance. Finally, importance of intact tumor cell-intrinsic IFN γ signaling in response/adaptive resistance to anti-PD1 therapy in

these models should be demonstrated, e.g. by experiments with IFN γ receptor knock-out cancer cells.

Discussion, statement by the authors: „.... PARP14 mRNA levels could not predict the depth or duration of clinical response among responders to ICBT in the Riaz et al. cohort (data not shown). This may reflect several confounding factors, including a limited number of samples available for analysis; ...“

Comment: Besides the Riaz cohort, additional data sets are available for analyses, e.g. Liu et al. (Nat. Med. 2019) and Gide et al. (Cancer Cell, 2019)

IFN γ signaling has strong anti-proliferative and pro-apoptotic activity in tumor cells, as shown in different mouse and patient models (Hoekstra et al., Nature Cancer 2020; Sucker et al., Nat Commun 2017). Does treatment of YUMM2.1, CT26 and MC38 with 50 U/ml affect tumor cell proliferation and survival? Does knock-down/out of PARP14 in tumor cells have similar effects when pretreatment is carried out with higher IFN γ concentrations?

Minor:

How was the 2-week IFN γ pre-treatment performed, was IFN γ added once or repeatedly?

Suppl. Fig. 2B: Binding of STAT1 to PARP14 promoter. Has CHIP-Seq been performed for STAT1 or pSTAT1, data for control K562 are missing.

Reviewer #3 (Remarks to the Author): with expertise in PARP14, cancer

This manuscript identifies PARP14 as a target to restore PD-1 immune checkpoint response in tumors showing adaptive resistance. In a series of beautiful experiments, the authors show that PARP14 genetic or pharmacological inhibition improves the response to anti-PD1 treatment of cells pre-treated with IFN, a pre-treatment which would otherwise make them not responsive to anti-PD1. The authors also document a change in the immune response upon PARP14 inhibition during this process, which may be responsible for or contribute to the observed phenotype (although this is not directly pursued by the authors). These are very important

findings, considering the clinical implications and the current relevance of PD-1 research.

This being said, the manuscript does not seek to identify any novel mechanisms, beyond exploring the previously-described functions of PARP14 in regulating the immune response. While the data implying PARP14 in the adaptation to anti-PD1 treatment is clear and the findings are important, this lack of mechanistic insights make this manuscript better fit to a more specialized journal.

Additional minor comments:

1. It is sometimes difficult to compare the data as it is shown split within different panels. For example, IgG control presented in Fig S1A-C needs to be included in the same graphs as the anti-PD1 data in Fig 1. Showing the controls in separate graphs in the supplementary data is not best practice.
2. Fig 2B: The experiment presented does not include the essential control of non-IFN treatment. IFN γ -treated shPARP14 samples should be compared directly to BSA-treated shNTC condition, to clarify to what extent PARP14 depletion overcomes the resistance induced by IFN γ treatment.
3. Fig 2E-F: is there an impact of PARP14 inhibition in the anti-PD1 response for tumors which were not pre-treated with IFN γ ?
4. Some type of statistical analyses should be included when showing the data presented in figures 1F-H, and wherever this type of data is presented in the manuscript.

We thank the reviewers for providing constructive criticism of our manuscript. Below (in red text interweaved with the reviewers' comments) is a point-by-point response to their comments, including the changes we have made for this revision.

Reviewer #1 (Remarks to the Author):

Both positive and negative roles of the IFN-g signaling in immunotherapy responses have been reported, which is likely determined by a multitude of factors, including the duration of IFN-g stimulation. To this end, the Minn group from UPenn reported that chronic stimulation with IFN-g induces immunotherapy resistance, via both PD-L1-dependent and -independent mechanisms (i.e., STAT1-mediated epigenetic changes). Given the pleiotropic functions of the IFN-g signaling, other underlying mechanisms likely exist. Here, Wong, et al. explored this further in an attempt to identify actionable targets for immunotherapy resistance resulting from chronic IFN-g exposure.

The reviewer nicely summarizes why further research in this area is required.

Using various tumor models for melanoma and colorectal cancer, they found that prolonged IFN-g pretreatment upregulated poly-ADP ribosyl polymerase 14 (PARP14) in tumor cells, coupled with anti-PD-1 resistance. Clinically, they found a positive correlation of PARP14 with IFNG in human skin cutaneous melanomas (SKCMs) from the TCGA database and that PARP14 expression was increased in patient samples treated with anti-PD-1. To test how PARP14 upregulation in tumor cells modulates anti-PD-1 resistance in tumors after chronic exposure to IFN-g, the authors knocked down PARP14 in tumor cells, which significantly improved mouse survival upon anti-PD-1. As a pharmacological approach, they treated mice bearing established IFN-g-pretreated tumors with RBN012759, an inhibitor for PARP14 (PARP14i). While PARP14i alone only suppressed YUMM2.1 melanoma but not CT26 or MC38 colorectal tumors, when combined with anti-PD-1, it resulted in potent suppression of all these tumors, with 100% of YUMM2.1-bearing mice as well as ~60% of mice bearing either CT26 or MC38 surviving to the end of experiments (60 days for melanoma, 55 days for CT26, and 35 days for MC38). Clearly, these results indicate that PARP14i resensitized these anti-PD-1-resistant tumors to anti-PD-1, leading to strong synergistic effects.

As a small correction of what the reviewer states, our results show that even for YUMM2.1, PARP14i alone is ineffective, but as the reviewer notes strongly synergizes with anti-PD-1 to control tumour growth in three syngeneic models.

To determine how PARP14 in host cells contributes to this resensitizing effect, the authors observed that PARP14 KO mice bearing YUMM2.1 melanoma responded better to anti-PD-1, suggesting host cell PARP14 is also important in the resensitization to anti-PD-1. They further demonstrated that CD8+ T cells were essential for therapeutic efficacy of PARP14i+anti-PD-1. While these data imply that PARP14 in both tumor and host cells regulates anti-tumor immunity and PARP14i boosted anti-PD-1 efficacy, overall data quality and significance need to be substantially improved.

Major Concerns

1. As aforementioned, immunotherapy resistance associated with chronic IFN-g treatment

has been reported by the Minn group, who also showed that direct inhibition of JAK1/2 with Ruxolitinib can overcome the immunotherapy resistance. This greatly dampens my enthusiasm and overall significance of this study.

Whether JAKi will turn out to be clinically useful in reversing adaptive resistance to immune checkpoint inhibitors is still being evaluated. However, the data in Benci et al. 2016 Cell 167, 1540-1554 imply that the window of opportunity to deploy JAKi needs careful definition, as administered too soon it has no effect on reversing aCTLA4 resistance (aPD1 resistance was not determined). Moreover, the dose of JAKi needs to be carefully titrated to antagonise IFN signalling only in tumour cells and not in immune effector cells. Furthermore, JAKi are myelo- and immunosuppressive and clinically associated with significant adverse effects (Lussana F, Cattaneo M, Rambaldi A, Squizzato A. Ruxolitinib-associated infections: A systematic review and meta-analysis. *Am J Hematol*. 2018 Mar;93(3):339-347. doi: 10.1002/ajh.24976) and may even promote lymphoma (Porpaczy E, et al. Aggressive B-cell lymphomas in patients with myelofibrosis receiving JAK1/2 inhibitor therapy. *Blood*. 2018 Aug 16;132(7):694-706. doi: 10.1182/blood-2017-10-810739). These are not concerns encountered with PARP14i in the clinic.

2. As demonstrated here and by the Minn group, chronic activation of the IFN-g signaling in tumor cells is detrimental to anti-PD-1 therapy. Why would further activation of IFN-g signaling with PARP14i be able to resensitize IFN-g-pretreated tumors to anti-PD-1? Also, this is opposite to previous reports showing JAK1/2i is effective at overcoming immunotherapy resistance in IFN-g chronically treated melanoma.

This is an excellent question and one not yet fully resolved as it will require work beyond the scope of the present study. However, for now, we infer that by increasing STAT1-driven antigen processing and presentation PARP14 inhibition increases the immunogenicity of tumour cells, while, as demonstrated by Mehrotra et al. (2013 J Allergy Clin Immunol. 131(2):521-31.e1-12, 2013) and Cho et al. (J Immunol. 191(6):3169-780, 2013) through characterisation of *parp14* KO mice, simultaneously favouring type 1 cell-mediated immune responses in the host. Indeed, we now show that PARP14i chronic pre-treatment of CD4+ and CD8+ rested T cells stimulated ex vivo (using CD3 + CD28 antibodies for reactivation) resulted in increased frequency of IFN γ and TNF α expressions and decreased frequency of LAP (TGF β) and IL-10 expressions (Figure 4F–I of the revised manuscript). Moreover, CD8+ T cells tended to express more Ki-67 marker, which indicated that PARP14i pre-treated CD8+ T cells appeared to be more proliferative (Figure 4J). Based on different studies, high expression of IFN γ and TNF α and low expressions of LAP and IL-10 in T cells is important in regulating tumour immunity. Combined, these observations may explain how PARP14i will result in better tumour control.

Despite decreasing STAT1 activity, systemic administration of JAKi can also be beneficial but only because at the low dose used in Benci et al. 2016, Cell 167, 1540-1554 (60 mg/Kg by IP injection for 5 consecutive days) it didn't suppress IFN signalling in immune cells. Whether such a fine balancing act can be achieved in patients would require clinical trials to determine.

3. The inconsistency of the models used, data reproducibility, and scientific rigor. For instance, naïve tumor cells without prolonged IFN-g pretreatment (Fig. 2A-C) were used in the PARP14KD and PAPP14 KO mouse experiments (Fig. 2G-I), but IFN-g pretreated tumor cells were used in the experiments with PAPP14i (Fig. 2D-E and supplemental Fig. 4)

For Fig 2A-C (and ED Fig 3) cells were indeed pre-treated with IFNg. This was clearly stated in the original manuscript text on page 6:

“We next implanted IFN γ pre-treated shNTC- or shPARP14-expressing YUMM2.1 and MC38 cells into mice and applied the same IgG2a or α -PD-1 treatment regimen described above (Fig. 2A).”

and also in the experimental design schematic (Fig 2A) and in the graph titles (Fig 2B and C, ED Fig 3B-E).

The reviewer is correct that IFN-g pre-treatment was not used to generate data for Fig 2G-I. This is because the point of this experiment (as explained on page 7 of the manuscript) was:

“To determine whether endogenous PARP14 expression by host cells could affect the overall efficacy of α -PD-1 therapy”

As, if not the case, this would have precluded using a drug to systemically suppress PARP14 activity. Besides confirming that PARP14 was not required for response to α -PD1, the experiment indicated that PARP14 activity in host cells might even be detrimental to α -PD1 efficacy, which is consistent with its well-known role in suppressing cellular immunity in

favour of humoral immunity (Mehrotra et al. 2013 *J Allergy Clin Immunol.* 131(2):521-31.e1-12; Cho et al. 2013 *J Immunol.* 191(6):3169-78). We don't see the value in re-running this experiment with IFN-g pre-treated cells, as we know we need to knock down PARP14 in such cells in order to resensitize to a-PD1.

also, different treatment regimens were used for the immunoprofiling studies in Fig. 3C and therapeutic studies in Fig. 2D;

The treatment regime (summarised in Fig. 3C) used to generate data in Fig. 3D was deliberately curtailed compared to Fig 2D-F as the goals were entirely different. For Fig 3D, we wanted to profile immune cells during the treatment response to see the effects of the PARP14i on the immune response for which we needed sufficient tumour material (it would not be possible to perform the analysis if the tumours had completely regressed or once the drug had left the system). For 2D-F we wanted to test the effect of PARP14i treatment on tumour growth and animal survival, so mice were treated and then followed at least until and sometimes beyond when control animals were removed from the study due to reaching the tumour burden endpoint.

In addition, different ending times were chosen for YUMM2.1 (60 days), CT26 (55 days), and MC38 (35 days).

The different endpoints used for the tumour models simply reflects the different tumour growth rates of the control treatment groups. 3Rs principles mean it is unethical to continue collecting data from animals beyond when the effect is demonstrated clearly (please see Prescott MJ, Lidster K (2017) Improving quality of science through better animal welfare: the NC3Rs strategy. *Lab Animal* 46(4):152-156. doi:10.1038/labana.1217).

All the experiments were done just once and not both sexes of mice were used, which is not compliant with the requirements of Nat. Comms.

The synergistic effect of a-PD1 and PARP14i (our main finding) was demonstrated for three independent tumour cell models which is clear replication of this result. Using multiple shRNA is also a form of independent replication. For the YUMM2.1 model the effect of

combination treatment on tumour growth has now been fully independently replicated in ED Fig 4F of the revised manuscript.

We have only used female mice because we need to re-batch mice at various time points in the different regimens both to make the work affordable but also so mice are not housed alone for extended periods (which causes stress) and this is not possible with male mice who fight to the death when introduced to non-littermates. These practices are norms for the field, mirroring the approaches used for example in Benci et al. (Cell 2016 167(6):1540-1554.e12. doi: 10.1016/j.cell.2016.11.022) and Williams et al. (Nat Commun. Jan 30;11(1):602, 2020. doi: 10.1038/s41467-020-14290-4). Indeed, we cannot find a single comparable study where male mice were used.

4. The causal relationship of PARP14 upregulation in tumor cells and host cells to anti-PD-1 resistance (or IFN-g resistance) remains to be established. As discussed above, IFN-g pretreated tumor cells (anti-PD-1-resistant) should be used in PARP14KD and PARP14 KO mouse experiments, like what they did with the PARP14i. PARP14 should be re-expressed in IFN-g-pretreated PARP14KD cells to establish a direct link of tumor PARP14 expression to anti-PD-1 resistance.

As stated above, we have used pre-treatment with IFN-g for KD cells and, as previously stated, disagree that this is valid for the KO experiment. Rescuing with RNAi-resistant PARP14 is technically very challenging as we (and others) find overexpression of PARP14 (a ~202 kDa protein) impossible to sustain in cell models. Using two independent shRNA was we feel a reasonable approach to confirm the connection between PARP14 and the resistance phenotype.

While their human data of a correlation between PARP14 and IFNG indicated that IFN-g (Fig. 1I) and anti-PD-1 (supplemental Fig. 7E) upregulated PARP14, they nevertheless didn't establish that PARP14 expression in human tumor cells drives resistance to either anti-PD-1 or IFN-g.

Again, it will be very challenging to test this in human cells at the genetic level as it is difficult to reconstitute the human immune system ex vivo with cell models manipulated this way. We are considering treating human tumour explants with PARP14i. However, we have not yet been able to secure biopsies from lesions that are radiographically confirmed to be progressing on treatment and displaying evidence of increased tumour intrinsic IFN signalling.

On an extended note, PARP14 should be knocked out with CRISPR-Cas9 to explicitly assess tumor PARP14 in anti-PD-1 resistance, unless this would be lethal to tumor cells.

Gene knockout using CRISPR is an orthogonal approach to RNAi (as is the approach we have in fact adopted, namely pharmacological antagonism with a highly selective inhibitor) but not necessarily a superior approach. We desired to deplete PARP14 to a level comparable to IFN-naïve cells, both because this is well tolerated by tumour cells but also because the PARP14 inhibitor we have in the clinic suppresses PARP14 but doesn't ablate it. Subsequently, we have tried to generate CRISPANTS for PARP14 but have been able to completely ablate in any of our cell models.

5. The relative importance of tumor vs host expression of PARP14 awaits further investigations. Their data support that PARP14KD in naïve (not IFN-g pretreated) tumor cells can prevent chronic IFN-g treatment-induced anti-PD-1 resistance, but they don't show that PARP14KD can remedy anti-PD-1 resistance after chronic IFN-g treatment.

As stated above, this is a mistaken belief.

On the other hand, deletion of CD8+ T cells completely abolished the efficacy of combined PARP14i+anti-PD-1 therapy in overcoming anti-PD-1 resistance in IFN-g-pretreated YUMM2.1 melanoma. This, together with the results from PARP14 KO mice, establishes an important role of PARP14 in CD8 T cells in PARP14i+anti-PD-1. Similar studies should be done with in vivo neutralizing antibodies against CD4+ T cells.

This is certainly a feasible experiment but not one we considered worthwhile to undertake as it would be difficult to interpret: CD4+ T cells comprise multiple subsets with diametrically contrasting roles in ICBT responses; thus while depleting Th1 cells is anticipated to strongly negate the effects of a-PD1, depleting Treg and Th2 cells would do the opposite.

Additionally, IFN-g-pretreated PARP14KO tumor cells should be used in Fig. 2A-C to explicitly pinpoint the role of tumor expression of PARP14 in this process.

Precisely what we have done.

Minor Concerns

1. What does “pretreatment” label on the X axis of Fig. 1B mean? Why is the label of the X axis in Fig. 1G different from that of Fig. 1F and 1H? Also, Fig. 2I was partially cut off.

We thank the reviewer for pointing out these anomalies. We will remove pretreatment from the X-axis of Fig. 1B. The time taken for the CT26 model to reach the treatment start size is more variable in our hands than for the other two models, which is why our origin is treatment start rather than day of inoculation. For consistency, though, we will edit all the graphs 1G-H to show days after treatment starts.

2. Please provide the rationale to study PARP14 but not PARP9 or PARP12, given they were all upregulated by IFN-g. Further, PARP9 should be examined, given the functional antagonism between PARP14 and PARP9.

We only have a clinical lead and in vivo tool compound for PARP14, so this study focuses on validating PARP14i as a combination treatment with a-PD1. Besides, existing art implicated PARP14 as an IFN antagonist, whereas, in contrast, PARP9 is a STAT1 activator (Bachmann et al. Mol Cancer. 2014 13:125. doi: 10.1186/1476-4598-13-125). In the revised manuscript, we included discussion about the possible role of PARP9 in this resistance context. There is scant prior art regarding PARP12, so we considered this to be beyond the scope of the present study.

3. It is surprising that no long-term protection was observed in tumor-bearing mice with their rigorous regimen of anti-PD-1 (300 ug/mouse for 4 doses), given that these tumors are regarded as responsive tumor types to immunotherapy.

We believe we have administered the drug correctly and presented our data without removing any mice. Indeed, for IFN-naïve models our data is highly comparable to published studies (Homet Moreno et al. 2016 Cancer Immunol Res. 4(10):845-857. doi: 10.1158/2326-6066.CIR-16-0060; Jin et al. 2022. Sci Rep 12, 3278 (2022). <https://doi-org.manchester.idm.oclc.org/10.1038/s41598-022-07153-z>)

4. Knock-down of PARP14 did not seem to work well (Supplemental Fig. 3A).

While the knockdown is not absolute, it generates a phenotype. RNAseq, corroborated by qPCR (ED FIG 3E) reveals at least a 40% significant reduction in *Parp14* mRNA in both cell lines – YUMM2.1 and MC38, after 2-week chronic IFN-g treatment.

In the case of YUMM2.1, we have also performed proteomic analysis which indicates ~70% reduction in protein using shRNA sequence 2 (data not shown) consistent with the western result in ED Fig 3E. Probably the majority of cells have greater > 50% knockdown and then a minority of cells have little to no knockdown, the latter appearing to be strongly selected in the MC38 model upon relapse, again consistent with the role of PARP14 in mediating resistance. The knockdown in IFN-treated cells is limited by the fact that *PARP14* is a STAT1 target but at the same time the gene product is a STAT1 antagonist. Therefore, as PARP14 protein (or activity in the case of the PAPER14i) falls, STAT1 activity increases together with *PARP14* mRNA abundance.

5. Increased expression of PARP14 after RBN012759 may obliterate the therapeutic effects of PARP14i. RBN012811 may be a better option.

Unfortunately, RBN012811 is not a good compound to test in vivo. Our clinical lead, RBN-3143, is a variant of RBN012759.

6. Fig. 5A showed upregulated PARP14 in anti-PD-1-resistant bulk tumor as compared to untreated bulk tumor. This should be compared to anti-PD-1-responsive bulk tumors. Also, analysis of PARP14 expression in tumor cells should be done.

In ED Fig 11C–D, we show there is no up-regulation of *Parp14* mRNA in on-treatment tumour samples compared to untreated tumours. By inference, therefore, since *Parp14* mRNA is upregulated in resistant tumours compared to untreated tumours (Fig 10A), but it is not significantly upregulated compared to responding tumours.

7. *Supplemental Fig. 7G was not mentioned in the text.*

Sorry, a small typo. *“The growth of tumours derived from IFN γ -naïve CT26 cells that regrew following α -PD-1 administration was also suppressed by PARP14i treatment (Extended Data Fig. 7F).”* should have ended *“...(Extended Data Fig. 7F, G).”*

8. *The gating strategies are not accurate.*

We have revised the gating used in the latest version of our manuscript.

9. *Both forward and reverse primer sequences should be provided in supplemental table 1.*

Our apologies, an oversight that will be corrected. Please see the Supplementary Table 1.

10. *“Depleting PARP14” and “depletion of PARP14” were often used, but only knock-down of PARP14 was performed.*

We feel that these terms are interchangeable with depleting/depletion being plain English and knockdown a jargon term.

In addition, the following sentences are confusing; please re-write.

“Contrasting IFNG and STAT1 mRNA abundance in pre- versus on-treatment melanoma biopsies also supported that IFN γ -signalling is induced by ICBT.”

“However, PARP14 mRNA levels could not predict the depth or duration of clinical response among responders to ICBT in the Riaz et al. cohort28 (data not shown). This may reflect several confounding factors, including a limited number of samples available for analysis; highly heterogeneous tumours in which only small regions were sampled; sampling of different lesions pre- and on-treatment; and finally, biopsies that were not necessarily sampled at the point of relapse, which is when our in vivo data indicate that PARP14 induction may be at its highest and exerting its greatest influence.”

In the revised manuscript, we provide more clarity on this issue by including data from a study from Lim and colleagues (*Nat Commun* **14**, 1516 (2023). <https://doi.org/10.1038/s41467-023-36979-y>) which show that in a significant fraction (6/18) of metastatic melanoma lesions progressing on ICBT tumour cell intrinsic IFN γ signalling and therein PARP14 expression are elevated.

Considering all these different factors, even a more robust longitudinal examination of PARP14 induction in patient biopsies undergoing α -PD-1 therapy may not provide a solution to the discrepancy between this study and the study by Riaz, et al., as it is not feasible to biopsy the tumor tissues from the same loci at different times.

We tend to agree with the reviewer as implied in our comment: *“...although this may be difficult to reconcile with patient care.”*

Reviewer #2 (Remarks to the Author):

Chronic IFN γ signaling in tumor cells has recently been linked to immunotherapy resistance. The underlying non-genetic mechanisms are still poorly understood but seem to be multifaceted. Here, Wong et al. propose involvement of PARP14 in this process. PARP14 (ARTD8) is known as a negative regulator of STAT1 and inflammatory genes, but findings of the authors in the context of chronic IFN γ signaling in tumor cells are novel and of potential clinical interest.

This reviewer appreciates the significant potential advance of our findings.

However, important data supporting their conclusions are missing.

Major concerns:

The authors summarize their findings in the discussion as follows: „... For our in vivo experiments, we employed YUMM2.1, CT26 and MC38 syngeneic mouse tumour models, which initially respond to α -PD-1 therapy but subsequently relapse, MC38 more quickly than YUMM2.1 or CT26. Chronic IFN γ treatment before implantation rendered tumours resistant from the outset to α -PD-1 therapy, but response to α -PD-1 could be restored by depleting PARP14 in tumour cells ...“

Comment: This is a general statement over all models on the role of tumor cell-intrinsic PARP14 in resistance to anti-PD1 therapy under chronic IFN γ exposure. However, in case of CT26, the role of tumor-intrinsic PARP14 has not at all been studied. According to the Western blot (suppl Fig. 1E/D, [labelling in the figure is not correct]), IFN γ -treated CT26 cells only barely upregulate PARP14, if at all. This raises concerns about the role of tumor cell-intrinsic PARP14 in anti-PD1 resistance of IFN γ -pretreated CT26 tumors. Does PARP14 CRISPR/Cas9 knock-out restore the anti-PD1 sensitivity of IFN γ -pretreated CT26 tumors? Can effects of PARP14i on IFN γ -pretreated CT26 tumors just be explained by its effects on host cells?

We provide improved western blots for CT26 showing a clear induction of PARP14 in Figure 2B of the revised manuscript. Unfortunately, CT26 expressing PARP14 KD vector were immunogenic and even control tumours rejected by host mice. Also, attempts to recover CT26 cells with complete ablation by CRISPR/Cas9 were unsuccessful.

shPARP14 YUMM2.1 and shPARP14 MC38 cells show only modest decrease in PARP14 protein expression (supp. Fig. 3a). The authors themselves define the effects of shPARP14 downregulation on anti-PD1 sensitivity of IFN γ -pretreated MC38 tumors as “less robust” (suppl Fig. 3D). In order to obtain clear results PARP14 should be knocked out (CRISPR/Cas9) in all models.

As discussed above, while the knockdown is not absolute, it generates a phenotype. RNAseq, corroborated by qPCR (ED FIG 3E), reveals at least a 40% reduction in *parp14* mRNA.

In the case of YUMM2.1, we have performed proteomic analysis which indicates ~70% reduction in protein using shRNA sequence 2 (data not shown) consistent with the western result in ED Fig 3A. Probably the majority of cells have greater > 50% knockdown and then a minority of cells have little to no knockdown, the latter appearing to be strongly selected in the MC38 model upon relapse, again consistent with the role of PARP14 in mediating resistance. The knockdown in IFN-treated cells is limited by the fact that *PARP14* is a STAT1 target but at the same time the gene product is a STAT1 antagonist. Therefore, as PARP14 protein (or activity in the case of the PAPP14i) falls, STAT1 activity increases together with PARP14 mRNA abundance.

Gene knockout using CRISPR is an orthogonal approach to RNAi (as is the approach we have in fact adopted, namely pharmacological antagonism with a highly selective inhibitor) but not necessarily a superior approach. We desired to deplete PARP14 to a level comparable to IFN-naïve cells, both because this is well tolerated by tumour cells but also because the PARP14 inhibitor we have in the clinic suppresses PARP14 but doesn't ablate it.

In Figure 5B/C the authors show that tumours which spontaneously relapsed following α -PD-1 therapy could be re-sensitised by PARP14i treatment. Does it have a clear impact also on survival?

We now demonstrate a survival advantage (Figure 10C in the revised manuscript).

Does combined anti-PD1 plus PARP14i treatment affect tumor growth in IFN γ -naïve models? It is expected that in anti-PD1 sensitive tumor models (YUMM2.1, CT26, MC38), IFN γ is released early after therapy onset, suggesting that early blockade of PARP14 activity by selective inhibitors should prevent or delay development of adaptive resistance.

We do not see a significant effect of PARP14i on IFN-naïve cell models treated with or without a-PD1 (see ED Fig 13A-C of the revised manuscript for the YUMM2.1 model and below for MC38 data).

RBN012759 CP MC38

This we believe is due to the very low base-line expression of PARP14 which remains low even following rounds of α-PD1 treatment (ED FIG 12C and D of the revised manuscript).

Indeed, we only see upregulation of *Parp14* in tumours that relapse following exposure to a-PD1 (Fig 9A of the revised manuscript).

Finally, importance of intact tumor cell-intrinsic IFN γ signaling in response/adaptive resistance to anti-PD1 therapy in these models should be demonstrated, e.g. by experiments with IFN γ receptor knock-out cancer cells.

Rather than IFNGR KO which may render cells inherently resistant to a-PD1, we demonstrated the ability of JAKi to antagonise relapse thus demonstrating a role for tumor cell-intrinsic IFN γ signalling in the spontaneous progressing model (Figure 10 of the revised manuscript).

Discussion, statement by the authors: „.... PARP14 mRNA levels could not predict the depth or duration of clinical response among responders to ICBT in the Riaz et al. cohort (data not shown). This may reflect several confounding factors, including a limited number of samples available for analysis; ...“

Comment: Besides the Riaz cohort, additional data sets are available for analyses, e.g. Liu et al. (Nat. Med. 2019) and Gide et al. (Cancer Cell, 2019)

We thank the reviewer for bringing this to our attention and have extended our analyses to other data sets. Unfortunately, none of them gave a significant result to predict any high PARP14 mRNA expression of patients having a shorter lifespan in any immunotherapy trials.

IFN γ signaling has strong anti-proliferative and pro-apoptotic activity in tumor cells, as shown in different mouse and patient models (Hoekstra et al., Nature Cancer 2020; Sucker et al., Nat Commun 2017). Does treatment of YUMM2.1, CT26 and MC38 with 50 U/ml affect tumor cell proliferation and survival? Does knock-down/out of PARP14 in tumor cells have

similar effects when pretreatment is carried out with higher IFN γ concentrations?

50 U/ml of IFN-g, equivalent approximately to ~2-4 ng/ml which reflects physiological IFN-g levels (Williams et al. Nat Commun. 2020 Jan 30;11(1):602. doi: 10.1038/s41467-020-14290-4.) reflected in the robust activation of STAT1, did not affect growth of these cells significantly. As we need to generate a lot of cells for mouse implantation experiments, we cannot use doses of IFN-g that induce senescence or death. Also, it is not obvious to us that these doses of IFN-g are experienced *in situ* by any but the few cells directly adjacent to sources of IFN-g secretion.

Minor:

How was the 2-week IFN γ pre-treatment performed, was IFN γ added once or repeatedly?

Apologies. This important detail was omitted but has been included in the revised manuscript. Treatment was performed as follows:

During the 2-week IFN γ pre-treatment, we firstly added IFN γ on day 0. When cells become 70-80% confluent, cells were passaged and IFN γ refreshed. It was repeated over 2 weeks.

Suppl. Fig. 2B: Binding of STAT1 to PARP14 promoter. Has CHIP-Seq been performed for STAT1 or pSTAT1, data for control K562 are missing.

Apologies. The upper track is missing its label and is indeed control K562 and has now been corrected. ChIP-seq was for STAT1 rather than pSTAT1.

Reviewer #3 (Remarks to the Author):

This manuscript identifies PARP14 as a target to restore PD-1 immune checkpoint response in tumors showing adaptative resistance. In a series of beautiful experiments, the authors show that PARP14 genetic or pharmacological inhibition improves the response to anti-PD1 treatment of cells pre-treated with IFN, a pre-treatment which would otherwise make them not responsive to anti-PD1. The authors also document a change in the immune response upon PARP14 inhibition during this process, which may be responsible for or contribute to the observed phenotype (although this is not directly pursued by the authors). These are very important findings, considering the clinical implications and the current relevance of PD-1 research.

This being said, the manuscript does not seek to identify any novel mechanisms, beyond exploring the previously-described functions of PARP14 in regulating the immune response. While the data implying PARP14 in the adaptation to anti-PD1 treatment is clear and the findings are important, this lack of mechanistic insights make this manuscript better fit to a more specialized journal.

It is disappointing that after so much high praise the reviewer concludes this. Potentially, there is not much more to discover about the molecular and cellular roles of PARP14 and the challenge now is applying this understanding as we have done to a practical and therapeutic end. We maintain that the demonstration that PARP14 mediates IFN-g dependent immune homeostasis and that this is now actionable—and may therefore have impact when translated to the clinic—is profound and will be of interest to a broad base.

Additional minor comments:

1. It is sometimes difficult to compare the data as it is shown split within different panels. For example, IgG control presented in Fig S1A-C needs to be included in the same graphs as the anti-PD1 data in Fig 1. Showing the controls in separate graphs in the supplementary data is not best practice.

We felt that this way of presenting the data best emphasized the profound effect of IFN-g pretreatment on a-PD1 response. The supplementary data simply established that IFN-g pretreatment has no significant effect on tumour development and growth. We have now though brought the control data into the main figure 1 to make the comparison easier.

2. Fig 2B: The experiment presented does not include the essential control of non-IFN treatment. IFN γ -treated shPARP14 samples should be compared directly to BSA-treated shNTC condition, to clarify to what extent PARP14 depletion overcomes the resistance induced by IFN γ treatment.

As PARP14 is barely expressed in IFN-naïve cells and we detected no significant growth difference in cells with PARP14 shRNA compared to control cells (consistent with the lack of effect of PARP14i on growth at pharmacologically relevant doses), we felt this would be a waste of mice. Moreover, there is no difference in the growth of tumours expressing shNTC or shPARP14 treated with IgG control antibody. Therefore, it is valid to conclude that the effect of PARP14 KD is specific to a-PD1 response and not a general effect of PARP14 KD on tumour development.

3. Fig 2E-F: is there an impact of PARP14 inhibition in the anti-PD1 response for tumors which were not pre-treated with IFN γ ?

As mentioned above for reviewer 2, we do not see a significant effect of PARP14i on IFN-naïve cell models treated with or without a-PD1 (see ED Fig 13A-C of the revised manuscript for YUMM2.1 model and below for MC38 data).

RBN012759 CP MC38

This we believe is due to the very low base-line expression of PARP14 which remains low even following rounds of α-PD1 treatment (ED FIG 12C and D of the revised manuscript).

Indeed, we only see upregulation of PARP14 in tumours that relapse following exposure to α-PD1 (Fig 9A of the revised manuscript).

4. Some type of statistical analyses should be included when showing the data presented in figures 1F-H, and wherever this type of data is presented in the manuscript.

Statistical significance was demonstrated for the change in tumour volume by the end of the treatment period as depicted in Fig 1B (also for other examples 2B, 2E, 5C), and is performed for average tumour growth curves elsewhere in the manuscript, as is typical practice.

REVIEWERS' COMMENTS

Reviewer #1 (Remarks to the Author):

Major Concerns

1. With respect to the KD of PARP14, I double-checked the original submission (Page 18, Method) and could not find the description that it was done in tumor cells pretreated IFN-g for two weeks. To be on the same page with the authors, "KD in naive tumor cells and then treated with IFN-g for two weeks" is different from "tumor cells treated with IFN-g for two weeks and then KD".

2. As the authors mentioned that chronic activation of the IFN-g signaling in tumor cells contributes to therapeutic resistance to anti-PD-1, why would further augmentation of IFN-g signaling by PARP14i be able to re-sensitize tumor cells with chronically activated IFN-g signaling and not responsive to further IFN-g stimulation (Fig. 2C-D) to anti-PD-1. If so, why would inhibition of the IFN-g signaling by Ruxolitinib be also effective? How to reconcile these two seemingly opposite findings?

3. Considering that PARP14i largely boosted effector cytokine production and concomitantly reduced Treg, in my opinion, blocking CD4 T cells should be done, as CD4 T cells provide essential help to CD8 T cells and may actually be the "real" driver of the therapeutic effects of PARP14i+anti-PD-1.

4. Regarding the effects of PARP14i on T cells (Fig. 4), justifications on how the regimen (stimulation, resting, and then re-stimulation) was decided and how closely relevant this would be to the in vivo situation (i.e., tumor-infiltrating T cells: TILs) should be provided.

Minor Concerns

1. Please clarify the discrepancy: in the text, it said "25% of mice bearing YUMM2.1 tumours treated with a combination of α -PD-1 and 500 mg/Kg PARP14i exhibited durable tumour regression (up to 60 days post-treatment) (Figure 3C)", but the data showed 100% mice could survive up to 60 days.

2. Please included a group of mice rechallenged with an unrelated tumor type to demonstrate the specificity of memory response (Fig. 3D).

3. Please clarify "Parp14 expression was significantly higher in Ifng-high tumours (ED Fig. 12E-F)" vs "Comparable gene expression changes were observed in short-term melanoma cell cultures derived from ICBT-progressing lesions with elevated IFN γ -signalling compared to cultures where intrinsic IFN-signalling was minimal⁷ (Fig. 9B)". These seem to be in direct conflict with each other.

4. Please specify the dose of Ruxolitinib used.

5. Please spell out abbreviations, when used for the first time (e.g. TCIRs).

Reviewer #2 (Remarks to the Author):

The authors addressed the points that I raised. Overall, the new data strengthen the role of PARP14 as potential therapeutic target in the context of anti-PD-1 resistance driven by chronic IFN γ exposure.

Reviewer #3 (Remarks to the Author):

In the revised manuscript, the authors satisfactorily addressed my comments and those of other reviewers on some of the technical aspects of their manuscript. In my original review, I had mentioned what was in my view the major weakness of this work, namely that it does not seek to identify novel mechanisms. While I still think this is a weakness, and I do not agree with the authors' response that "there is not much more to discover about the molecular and cellular roles of PARP14", I do agree with their statement that applying the current knowledge of PARP14 functions to a therapeutic end is very important. In this context, I do believe that their manuscript is indeed a worthy addition to the literature.

We thank the reviewers for providing constructive criticism of our manuscript. Below (in red text interweaved with the reviewers' comments) is a point-by-point response to their comments, including the changes we have made for this revision.

Reviewer #1 (Remarks to the Author)

Major Concerns

1. With respect to the KD of PARP14, I double-checked the original submission (Page 18, Method) and could not find the description that it was done in tumor cells pretreated IFN-g for two weeks. To be on the same page with the authors, "KD in naive tumor cells and then treated with IFN-g for two weeks" is different from "tumor cells treated with IFN-g for two weeks and then KD".

We apologise for any unintended ambiguity in the original text. In the text of the revised manuscript, we have made it clear enough, we feel, that we first created cell models in which endogenous *Parp14* expression was permanently suppressed by the expression of shRNA targeting *Parp14* mRNA and that these stable cell models along with control cells expressing a non-targeting shRNA were then treated with IFN-g for two weeks before implantation in mice.

2. As the authors mentioned that chronic activation of the IFN-g signaling in tumor cells contributes to therapeutic resistance to anti-PD-1, why would further augmentation of IFN-g signaling by PARP14i be able to re-sensitize tumor cells with chronically activated IFN-g signaling and not responsive to further IFN-g stimulation (Fig. 2C-D) to anti-PD-1. If so, why would inhibition of the IFN-g signaling by Ruxolitinib be also effective? How to reconcile these two seemingly opposite findings?

The reviewer highlights an interesting paradox that we cannot currently answer definitively without significant further experimental work which will be the focus of our subsequent enquiries. At present, we can only speculate that while the Minn group has shown that suppressing STAT1 activity selectively in tumour cells through low-dose and short-term JAK inhibitor treatment reinstates anti-PD/L1 sensitivity through suppressing TCIR expression among other immunosuppressive pathways, greatly augmenting STAT1 activity in both host and tumour cells responding to IFN-g by PARP14 inhibitor also reinstates anti-PD/L1 sensitivity despite activating STAT1, presumably through a STAT1-driven increase in tumour cell immunogenicity (that is by boosting antigen expression, processing and presentation in both tumour cells and host cells), thereby counteracting the effects of any immunosuppressive molecules that are simultaneously induced. In summary, the immunosuppression resulting from chronic activation of STAT1 in tumour cells (resulting in turn from chronic IFN-g exposure) can be reset either through downregulation of JAK-STAT1 signalling or through pushing STAT1 activity to a higher point (including in host APCs) at which it re-instates tumour cell immunogenicity. Goldilocks effects—wherein too little or too much of a determinant confers reduced tumour cell fitness—has been described for various transcription factors and possibly reflects divergent gene expression programs and therein phenotypic effects being realised at different levels of promoter occupancy.

3. Considering that PARP14i largely boosted effector cytokine production and concomitantly reduced Treg, in my opinion, blocking CD4 T cells should be done, as CD4 T cells provide essential help to CD8 T cells and may actually be the "real" driver of the therapeutic effects of PARP14i+anti-PD-1.

While we share the reviewer's enthusiasm for exploring the contribution of CD4 T cells to the therapeutic effects of PARP14i+anti-PD-1, we don't agree that this is best achieved through globally depleting CD4 cells. As we previously argued, CD4 cells encompass a great many subtypes with pro-and anti-tumour effects, and it will not be possible to interpret the outcome of depleting them all (that is the outcome will reflect down-regulating both pro-tumourigenic and anti-tumourigenic effects of CD4 cells).

4. Regarding the effects of PARP14i on T cells (Fig. 4), justifications on how the regimen (stimulation, resting, and then re-stimulation) was decided and how closely relevant this would be to the in vivo situation (i.e., tumor-infiltrating T cells: TILs) should be provided.

Given the importance of CD4 cells in shaping an immune response following stimulation alluded to already by the reviewer and the documented effects of PARP14 in promoting type 2 immune responses (which are widely viewed as permissive to tumour development), we wanted to demonstrate that using a PARP14 inhibitor would promote the development of a type 1 immune response (required for tumour clearance). Also, we suspected that PARP14 is important for transcription factors like STAT1 to mediate durable epigenetic remodelling of gene expression which may require multiple cell cycles to achieve. For this reason, we focused on the ability of chronic exposure to PARP14i of T cells forced to proliferate through TCR stimulation to spontaneously differentiate toward type 1, by evaluating the expression of type 1 cytokines IFN γ and TNF- α by both CD4 and CD8 T cells. As anticipated, we observed significant increases in the percentages of IFN γ and TNF- α -producing CD4 and CD8 T cells in cells treated chronically with RBN012759 to antagonize PARP14 (Figure 4F–G; Extended Data Fig. 5 for gating strategy). Conversely, significant decreases in latency associated peptide (LAP)+ (latent TGF β), and IL-10+ CD4 and CD8 T cells were also seen (Figure 4H–I). Interestingly, PARP14i-pre-treated CD8+ T cells appeared to be more proliferative expressing a higher percentage of Ki-67+ (Figure 4J; Extended Data Fig. 6 for gating strategy). We concluded that PARP14i treatment could induce a proinflammatory type 1 phenotype in both CD4+ and CD8+ T cells.

Minor Concerns

1. Please clarify the discrepancy: in the text, it said "25% of mice bearing YUMM2.1 tumours treated with a combination of α -PD-1 and 500 mg/Kg PARP14i exhibited durable tumour regression (up to 60 days post-treatment) (Figure 3C)", but the data showed 100% mice could survive up to 60 days.

We don't see a discrepancy here: 25% of mice bearing YUMM2.1 tumours treated with a combination of α -PD-1 and 500 mg/Kg PARP14i exhibited durable tumour regression (up to 60 days post-treatment), that is they showed a complete response. Overall, 100% of mice bearing YUMM2.1 tumours treated with a combination of α -PD-1 and 500 mg/Kg PARP14i lived up to 60 days of post-tumour implantation (Figure 3C). Thus, a further 75% of mice could survive at least up to 60 days as well as those with a complete response because their tumour

growth was still significantly suppressed, this is they have stable disease or slowly progressing disease.

2. Please included a group of mice rechallenged with an unrelated tumor type to demonstrate the specificity of memory response (Fig. 3D).

While this is practicable to do, it will cost a significant time delay (at least an extra 4 months) and resource investment but will not change the conclusion at all. Indeed, it would be highly unexpected (miraculous really) if protection were conferred to an unrelated tumour type.

3. Please clarify "Parp14 expression was significantly higher in Ifng-high tumours (ED Fig. 12E-F)" vs "Comparable gene expression changes were observed in short-term melanoma cell cultures derived from ICBT-progressing lesions with elevated IFN γ -signalling compared to cultures where intrinsic IFN-signalling was minimal⁷ (Fig. 9B)". These seem to be in direct conflict with each other.

Again, we don't see a discrepancy: *Parp14* mRNA expression in our murine YUMM2.1 and MC38 tumour models correlates with *Ifng* mRNA expression (independently of anti-PD1 treatment). This mirrored the pattern in ICBT-progressing lesions where *PARP14* mRNA expression again correlates with *IFNG* mRNA expression.

4. Please specify the dose of Ruxolitinib used.

Apologies. An oversight. In the Methods of our revised manuscript we state that 'Mice were administered vehicle or 60 mg/Kg of Ruxolitinib (MedChemExpress) by i.p. injection once a day (QD). Ruxolitinib was dissolved in 0.5% methylcellulose (Sigma-Aldrich) + 0.1% Tween 80 (Sigma-Aldrich). Each dose was delivered in a volume of 0.1 mL/20 g mouse (10 mL/kg) and adjusted for the last recorded weight of individual animals'.

5. Please spell out abbreviations, when used for the first time (e.g. TCIRs).

Apologies. All abbreviations will be spelled out at first use in our next revision, although it is difficult to judge in every case what is a widely known acronym (e.g. DNA) and what more specialist.

Reviewer #2 (Remarks to the Author)

The authors addressed the points that I raised. Overall, the new data strengthen the role of PARP14 as potential therapeutic target in the context of anti-PD-1 resistance driven by chronic IFN γ exposure.

We thank the reviewer for appreciating the significant potential advance of our findings.

Reviewer #3 (Remarks to the Author)

In the revised manuscript, the authors satisfactorily addressed my comments and those of other reviewers on some of the technical aspects of their manuscript. In my original review, I had mentioned what was in my view the major weakness of this work, namely that it does not seek to identify novel mechanisms. While I still think this is a weakness, and I do not agree with the authors' response that "there is not much more to discover about the molecular and cellular roles of PARP14", I do agree with their statement that applying the current knowledge of PARP14 functions to a therapeutic end is very important. In this context, I do believe that their manuscript is indeed a worthy addition to the literature.

We thank the reviewer for appreciating the significant potential advance of our findings.